# MHC-I upregulation by macbecin II in the solid tumors potentiates the effect of active immunotherapy

Ravindra Pramod Deshpande [iD][1], Kerui Wu[2], Shih-Ying Wu[1], Abhishek Tyagi [iD][1], Eleanor C Smith [iD][1], Jee-Won Kim[1] & Kounosuke Watabe [iD][1✉]

## Abstract

**We aimed to restore MHC-I expression on the surface of solid tumors including breast cancer and melanoma cells to regain sensitivity to immunotherapy and suppress metastatic progression. We screened a natural compound library and identified macbecin II as a reagent that upregulates MHC-I expression and induces antigen-dependent cell death in pre-invasive and invasive breast cancer models. Furthermore, we employed active immunotherapy using engineered small extracellular vesicles from dendritic cells (DCs) as a tumor vaccine (IL2-ep13nsEV) in combination with macbecin II for personalized breast cancer treatment. We found that macbecin II induced MHC-I-dependent antigen presentation and that IL2-ep13nsEV synergized with macbecin II inducing cell death, reducing metastasis, and boosting immune cell infiltration. In addition, macbecin II potentiated the effects of anti-PD-1 immunotherapy in suppressing tumor growth and metastasis. Mechanistically, macbecin II upregulated MHC-I expression post-translationally by rescuing it from lysosomal degradation. Our findings revealed a strong efficacy of macbecin II in regulating MHC-I expression and following antigen-dependent cell death. Therefore, combining active immunotherapies and macbecin II represents an effective strategy to prevent growth and progression of solid tumors including breast cancer and melanoma.**

**Keywords** Active Immunotherapy; Anti-PD-1; Breast Cancer; Macbecin II; MHC-I

**Subject Categories** Cancer; Immunology

## Introduction

The MHC-I complex consists of heterodimers of polymorphic heavy and light chains (Blees et al, 2017). It presents peptides to CD8 effector cells of the immune system, resulting in the death of cancer cells (Sykulev, 2023). This process serves as the first step for anticancer immunity, and any malfunction or abnormal regulation of this process may result in the escape of tumor cells from immune surveillance (Wu et al, 2023b). Although the recent development of novel immunotherapies has revolutionized cancer treatment, most breast cancers display resistance to immune checkpoint blockade and engineered chimeric antigen receptor T (CAR-T) immunotherapies (Adams et al, 2019; Deshpande et al, 2020; Somboonpatarakun et al, 2024). A possible reason is that breast tumors often have a low mutational burden compared with other cancers (Castle et al, 2019). Furthermore, patients with breast cancer often have impaired antigen presentation, possibly due to defective co-stimulatory molecules (Wolfram et al, 2000). In addition, loss of MHC-I is described as a mechanism of acquired as well as intrinsic resistance to immunotherapy in cancer patients (Taylor and Balko, 2022). MHC-I expression is downregulated in 40–90% of cancer patients, including those with breast, lung, prostate, and melanoma (Cornel et al, 2020; Dhatchinamoorthy et al, 2021). A study in breast cancer (Kaneko et al, 2011) has shown that MHC-I downregulation is associated with increased lymph node metastasis, TNM stage, and worse disease-free survival, as well as with poor prognostic outcomes in several cancer subtypes, including colorectal (Anderson et al, 2021), non-small cell lung cancer (NSCLC) (Montesion et al, 2021), and breast (Dersh and Yewdell, 2021) cancer. Multiple studies have shown that increased MHC-I expression correlates with more tumor-infiltrating lymphocytes (TILs) in breast (Park et al, 2019) and colorectal cancers (Na et al, 2021; Anderson et al, 2021; Anderson et al, 2021). Therefore, the intrinsic reversible nature of MHC-I expression provides a mechanism to restore antigen presentation and accelerate adaptive antitumor immunity (Cornel et al, 2020). Consequently, restoring MHC-I expression is an attractive strategy to sensitize breast cancer patients to immunotherapies.

Ductal Carcinoma in Situ (DCIS) is the pre-invasive stage of breast cancer during which breast epithelial cells undergo abnormal proliferation and growth but remain confined within the mammary ducts (Allred, 2010). Sixty-two percent of hormone receptor-negative DCIS patients were found to have under expression of MHC-I, which may contribute to its progression to invasive breast cancer (Han et al, 2022). DCIS accounts for 20–30% of all breast cancers, with 40% of DCIS lesions reported to progress to invasive

[1]Department of Cancer Biology, Wake Forest University School of Medicine, Winston-Salem, NC 27157, USA. [2]University of North Carolina, Greensboro, NC 27412, USA. ✉E-mail: kwatabe@wakehealth.edu

breast cancer if left untreated (McCormick et al, 2015; Poelhekken et al, 2023). Treatment for DCIS typically involves surgical resection (mastectomy and lumpectomy), followed by radiotherapy and hormone deprivation for ER-positive cases (Carraro et al, 2014; Co et al, 2021). Despite these treatment regimens, it remains largely unknown which types of DCIS will progress to the invasive form. Therefore, treating DCIS patients with an active immunotherapy such as tumor vaccine to prevent recurrence and metastasis represents an attractive alternative strategy. We have previously demonstrated that engineered dendritic cell (DC)-derived small extracellular vesicles (sEVs) can induce robust anticancer immunity (Wu et al, 2023a). However, low expression of MHC-I in DCIS may compromise the efficacy of such immunotherapy. In the present study, we screened a natural compound library and identified macbecin II which upregulated MHC-I expression and enhanced antigen-dependent killing of cancer cells. Therefore, we examined the efficacy of macbecin II in synergizing with active immunotherapies, including vaccines and immune checkpoint blockade in our animal models. The combination treatment of macbecin II and active immunotherapies presented here has the potential to improve the management of early and late-stage breast cancer and mitigate disease progression.

## Results

### Screening of a natural compound library to identify molecule stimulating MHC-I expression

MHC-I expression is often downregulated in DCIS to evade immune surveillance (Han et al, 2022), which is known to contribute to the malignant progression of breast cancer (Garrido et al, 2018). Therefore, we aimed to identify compounds that upregulate MHC-I expression in DCIS.com cells using a natural compound library (Fig. 1A). The results of the initial screening indicated that macbecin II, cassythicine, and baccatin III significantly upregulate MHC-I (Fig. 1B). We confirmed these results by examining the expression of cell surface MHC-I in DCIS.com cells following treatment with the three identified compounds and found that macbecin II consistently upregulated MHC-I compared to cassythicine and baccatin III (Fig. 1C). Furthermore, we treated DCIS.com cells with a low dose of macbecin II (0.1 and 0.5 μM) and found that macbecin II significantly increased cell surface MHC-I expression even at the low dose (Fig. 1D). We also observed a similar effect of macbecin II on the invasive breast cancer cell line MCF10CA1a (Fig. 1E), as well as on murine triple-negative cancer cell lines E0771 (Fig. EV1A) and 4T1 (Fig. EV1B). Similar to MHC-I, macbecin II treatment was found to upregulate MHC-II expression on the surface of DCIS.com (Fig. EV1C) and MCF10CA1a (Fig. EV1D) cells. In addition, macbecin II was found to upregulate MHC-I protein expression in DCIS.com and MCF10CA1a cells (Fig. 1F,G). Together, these results demonstrate that macbecin II promotes MHC-I and II expression in breast cancer cells.

### Macbecin II promotes antigen presentation and enhances antigen-dependent cancer cell death

The expression of MHC-I on a cells surface is essential for antigen presentation and subsequent recognition by effector cells of the immune system (Handoko et al, 2024). To examine the potential of macbecin II in eliciting an antigen-specific CD8 T cell response, we used an ovalbumin (Ova) model system and found that treatment of B16 cells with macbecin II significantly enhanced H-2Kb Ova presentation on the cell surface (Fig. 2A). To clarify the role of macbecin II in antigen presentation during T cell activation, DCs derived from mouse bone marrow (BMDCs) were pulsed with Ova in the presence of macbecin II. We observed a significant enhancement in antigen presentation in the BMDCs with macbecin II treatment (Fig. 2B). In addition, macbecin II treatment enhanced MHC-I expression on the cell surface of BMDCs (Fig. EV2A). Subsequently, BMDCs were co-cultured with OT-1 T cells to examine T cell activation. Activation of OT-1 T cells was assessed by measuring IFN-γ expression with FACS (Fig. 2C) and TNF-α levels (Fig. EV2B) by ELISA. Our results demonstrated significant activation of OT-1 T cells when they were co-cultured with ova-pulsed DCs. In addition, macbecin II was found to significantly enhance lymphocyte activation marker CD69 expression on OT-1 T cells at Day 3 (Fig. EV2C). Furthermore, we evaluated the efficacy of activated OT-1 cells in mediating antigen-dependent killing of target cells by co-culturing OT-1 T cells with B16-Ova cells in the presence of macbecin II. Our results revealed that OT-1 T cells exerted a significant cytotoxic effect on B16-Ova cells, which was dependent on MHC-I expression (Fig. 2D). To further confirm the role of macbecin II in inducing antigen-dependent cell death, we first ectopically expressed ova in murine breast cancer cells (E0771) and then treated with macbecin II. We found that macbecin II induced the cell surface expression of H-2Kb-Ova (Fig. EV2D). The BMDCs were isolated from spleen of syngeneic mouse and pulsed with ova peptide. The ova-pulsed BMDCs were then treated with or without macbecin II and used for co-culture with the T cells. TNF-α secretion was examined in the supernatant as a marker of T cell activation. We found that the ova-pulsed and macbecin II-treated BMDCs significantly promoted TNF-α secretion from the T cells (Fig. EV2E). In addition, we co-cultured these T cells with E0771-Ova cells in the presence of macbecin II and found that they significantly reduced tumor cell viability (Fig. EV2F). These results demonstrate that macbecin II effectively promotes antigen presentation and the killing of cancer cells.

### Macbecin II potentiates anticancer efficacy of IL2-ep13nsEV tumor vaccine

Breast cancer patients often show resistance to immunotherapies (Michaels et al, 2024) due to reduced antigen presentation (Dhatchinamoorthy et al, 2021) and lower mutation burden (O'Meara and Tolaney, 2021). Considering these limitations, we previously demonstrated that personalization (p13n) of cancer treatment using dendritic cell-derived small extracellular vesicles (sEVs) as a vaccine was an effective approach to induce a strong antitumor response (Wu et al, 2023a). The function of the sEV vaccine is further enhanced by the expression of IL2 on sEVs, which boosts T cell activation. In addition, lipopolysaccharides (LPS) and stimulator of interferon genes (STING) agonist C-diGMP were shown to enhance the expression of co-stimulatory factors on sEVs. This enhanced version of active immunotherapy (IL-2-ep13nsEV) was shown to activate T cells without conventional antigen presentation by APCs (Wu et al, 2023a). To test the efficacy of macbecin II on this vaccine, we first

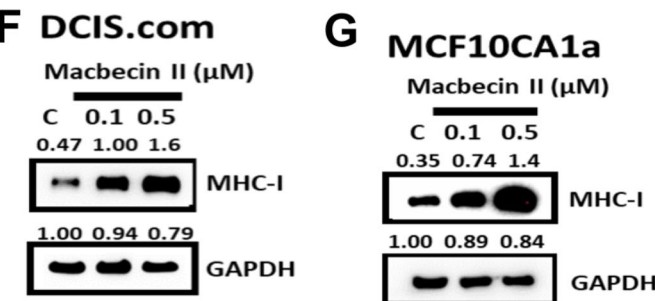

**Figure 1. Screening of a natural compound library to identify molecule stimulating MHC-I expression.**

(**A**) DCIS.com cells were treated with the natural compound library (5 μM) for 48 h. The cells were washed with PBS and stained with anti-HLA ABC primary and secondary antibodies. Readouts were obtained at 480 nm. (**B**) Fold change was calculated by comparing the MHC-I expression in DCIS.com cells treated with macbecin II and the empty vehicle. (**C**) DCIS.com cells were treated with the compounds listed in (**B**) at the dose of 5 μM for 48 h. Cells were trypsinized and stained with anti-HLA primary and secondary antibodies. The readouts were obtained by a BD FACS Canto. Mean fluorescence intensity was calculated and compared using one-way ANOVA with a Tukey post-hoc test ($n = 3$/group, biological replicates). (**D**) DCIS.com cells were treated with macbecin II at the indicated dose for 48 h. Post-incubation, cells were processed as described in (**C**) and compared using one-way ANOVA with a Tukey post-hoc test ($n = 3$/group, biological replicates). (**E**) MCF10CA1a cells were treated with macbecin II at the indicated dose for 48 h. Cells were processed as described in (**C**) and compared using one-way ANOVA with a Tukey post-hoc test ($n = 3$/group, biological replicates). (**F, G**) DCIS.com (**F**) and MCF10CA1a (**G**) cells were treated with the indicated dose of macbecin II for 48 h. Post-incubation, the total protein was isolated and MHC-I expression was examined by western blot. Data are represented as mean $+/-$ SEM. Source data are available online for this figure.

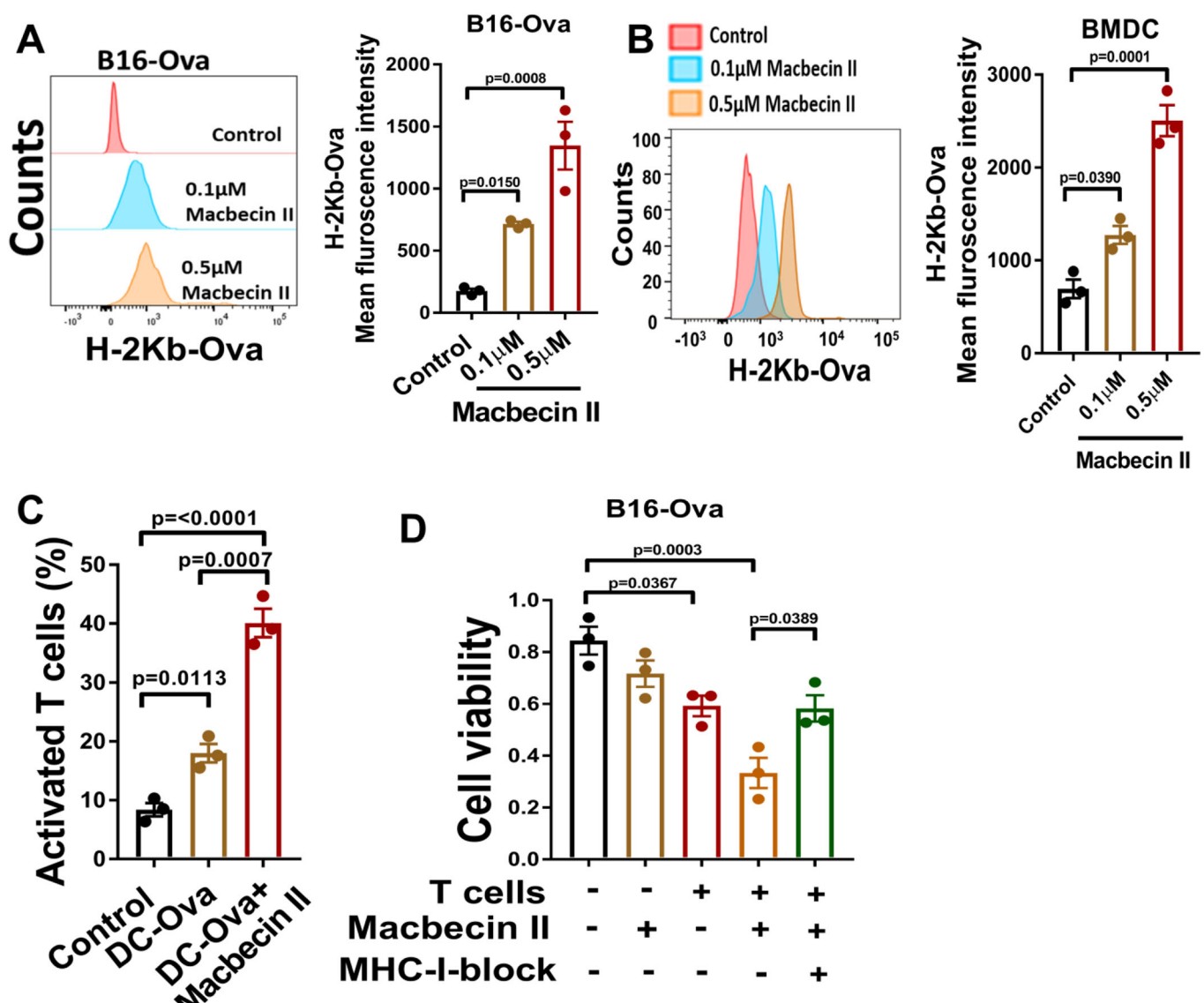

**Figure 2. Macbecin II promotes antigen presentation and enhances antigen-dependent cancer cell death.**

(A) B16-Ova cells were treated with the indicated dose of macbecin II for 48 h and the cells were examined for H-2kb-Ova peptide presentation by FACS. One-way ANOVA with a Tukey post-hoc test was used for comparison (n = 3/group, biological replicates). (B) BMDCs were isolated from the tibia of a C57BL/6 mouse. Dendritic cells (DC) were differentiated with GM-CSF treatment for 5 days. Loosely adherent cells were then isolated and treated with Ova (100 μg/mL) overnight. The next day, the cells were treated with the indicated dose of macbecin II for 48 h. Post-incubation, cells were washed and H-2Kb-Ova presentation was examined by FACS. Mean fluorescence intensity was calculated and compared by one-way ANOVA with a Tukey post-hoc test (n = 3/group, biological replicates). (C) The OT-1 T cells were co-cultured with DC in the presence of Macbecin II or PBS, and they were examined for IFN-γ expression by FACS as described in (B). Statistical significance was analyzed by the one-way ANOVA with a Tukey post-hoc test (n = 3/group, biological replicates). (D) B16-Ova cells (500 cells/well) were seeded into a 96-well plate and treated with the macbecin II (0.5 μM), OT-1 T cells (E:T ratio 10:1), and MHC-I blocking antibody (40 μg/mL) for 48 h. Dead cells and OT-1 T cells were washed off with PBS, and the remaining cancer cells were fixed with methanol and stained with crystal violet. The dye was then dissolved in 10% acetic acid, and the absorbance was measured at 590 nm. Statistical significance between groups was determined using the one-way ANOVA with a Tukey post-hoc test (n = 3/group, biological replicates). Data are presented as mean +/−SEM. Data are represented as mean +/− SEM. Source data are available online for this figure.

isolated monocyte-polarized DCs from HLA-matched PBMCs and pulsed them with DCIS.com lysate followed by expression of IL2-MFGE8 and treatment with C-diGMP and LPS. sEVs were then isolated and co-cultured with PBMCs (Fig. 3A). We confirmed the size and distribution of extracellular vesicles by nanoparticle tracking and electron microscopy (Fig. EV3A,B). In addition, we showed that the exosome enriched fraction activated the T cells more significantly as compared with the exosome-depleted supernatant (Fig. EV3C,D). CD8 T cells among the PBMCs were evaluated for activation by INF-γ, and our results showed that the PBMCs co-cultured with IL-2-ep13nsEV were significantly enriched with activated CD8 T cells and the macbecin II-treated tumor cells have the superior capacity to induce the INF-γ secretion as compared with the non-treated cells (Fig. 3B). The PBMCs were

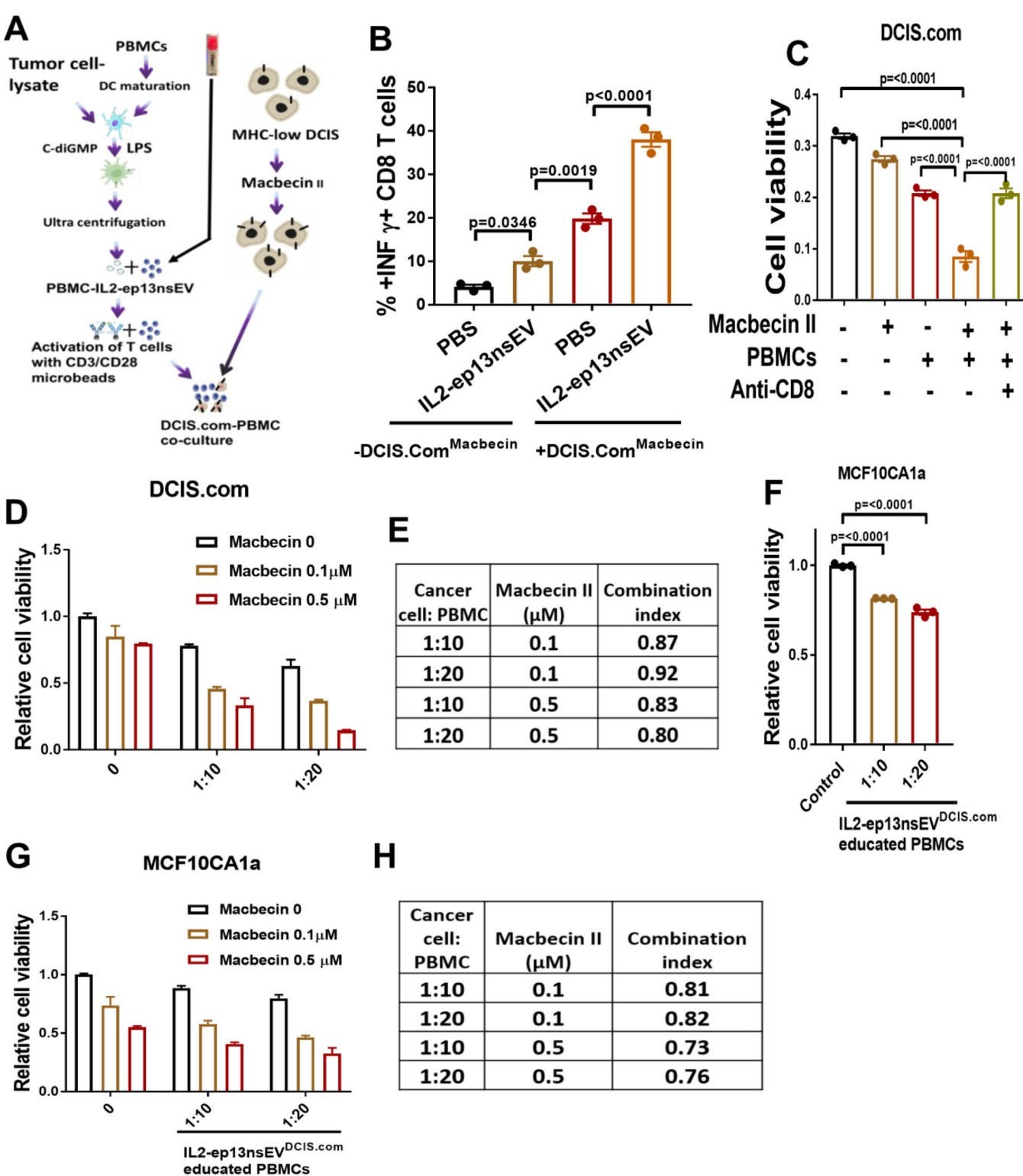

then co-cultured with DCIS.com cells in the presence of macbecin II to examine cell viability. We found that macbecin II significantly promoted tumor cell death, and this cell killing effect was suppressed when CD8 T cells were blocked (Fig. 3C). Similar to bocking the CD8 T cells, inhibiting the MHC-II rescued the cell death of DCIS.com cells when they were co-cultured with IL2-ep13nsEV-educated PBMCs in the presence of macbecin II (Fig. EV3E). To confirm the specific activation of T cells by the vaccine, we first pulsed BMDCs that were isolated from C57BL/6 mice with OVA peptide, with or without macbecin II. Next, IL2-

**Figure 3.  Macbecin II potentiates anticancer efficacy of IL2-ep13nsEV tumor vaccine.**

(A) Schematic of the experiment. HLA-matched PBMCs were treated with GM-CSF and IL-4 for 5 days to differentiate monocytes into dendritic cells (DC). The loosely adherent cells were isolated and treated with DCIS.com lysate, LPS (100 ng/mL), and C-diGMP (200 μM). Exosomes were then isolated by ultracentrifugation and co-cultured with HLA-matched PBMCs (20 μg/mL) for 5 days. The educated PBMCs were then expanded with CD3/CD28 microbeads and used for co-culturing with DCIS.com cells. (B) PBMCs from (A) were co-cultured with the DCIS.Com cells treated with or without macbecin II (0.5 μM). Followed by this, IFN-γ + CD8 T cell were examined by FACS. Statistical significance was analyzed by the one-way ANOVA with a Tukey post-hoc test ($n = 3$/group, biological replicates). (C) DCIS.com cells (500 cells/well, $n = 3$/group, biological replicates) were seeded in a 96-well plate and co-cultured in the presence of educated PBMCs, the indicated dose of macbecin II, or 40 μg/mL CD8 blocking antibody for 48 h. Post-incubation, cells were washed with ice-cold PBS to dislodge the dead cells. The live cells were fixed with methanol at room temperature for 15 min and stained with crystal violet. The dye was dissolved in 10% acetic acid and readouts were obtained at 590 nm. Statistical significance was compared by one-way ANOVA with a Tukey post-hoc test. (D) DCIS.com cells (500 cells/well) were cultured in a 96-well plate and treated with (i) vehicle (ii) macbecin II (0.1 μM and 0.5 μM), (iii) PBMCs (1:10 and 1:20 ratio) alone or in combination for 48 h ($n = 3$/group, biological replicates). After incubation, the cells were washed with ice-cold PBS to remove dead cells and PBMCs. Surviving cells were fixed in methanol and stained with crystal violet. The dye was dissolved in 10% acetic acid and readouts were obtained at 590 nm. (E) Combination index (CI) values were calculated by CompuSyn based on the inhibitory effect of macbecin II and PBMC treatment. A CI < 1 indicates synergy. (F) MCF10CA1a cells (1000 cells/well, $n = 3$/group, biological replicates) were seeded in a 96-well plate and treated with the indicated ratio of PBMCs for 48 h. Post-incubation, relative cell viability was calculated by the crystal violet assay method as mentioned in (C). Statistical significance was determined by one-way ANOVA with a Tukey post-hoc test. (G, H) MCF10CA1a cells (500 cells/well, $n = 3$/group, biological replicates) were seeded in a 96-well plate and co-cultured in the presence of educated PBMCs in combination with the indicated dose of macbecin II or 40 μg/mL CD8 antibody for 48 h. The co-culture experiment was performed as mentioned in (C). Combination index (CI) values were calculated by CompuSyn based on the inhibitory effect of macbecin II and PBMC treatment. A CI < 1 indicates synergy. (H) Data are represented as mean +/− SEM. Source data are available online for this figure.

ep13sEV were prepared from BMDCs by ultracentrifugation. Subsequently, splenocytes were harvested from syngeneic mice that were inoculated with OVA peptide and treated with or without macbecin II. These splenocytes were then co-cultured with IL2-ep13sEV. Finally, CD8 T cells from the co-culture were examined for tetramer-positive cells (Fig. EV3F,G). We found that IL2-ep13sEV significantly enhanced antigen presentation in the presence of macbecin II. In addition, we examined whether MHC expression on dendritic cells or cancer cells was required for a better response. We isolated bone marrow-derived dendritic cells (BMDCs) from syngeneic mice. The BMDCs were pulsed with B16-F1 cell lysate in the presence or absence of macbecin II. BMDCs treated with or without macbecin II and lysate were then co-cultured with T cells isolated from the spleen of syngeneic mouse. T cells co-cultured with macbecin II-treated, lysate-pulsed BMDCs were designated as T+, while those co-cultured with macbecin II-untreated BMDCs were designated as T−. Subsequently, B16-F1 cells were treated with or without macbecin II to upregulate MHC-I expression, followed by co-culture with the T cells. We found that MHC-I expression on both dendritic and cancer cells is essential for the effector function (Fig. EV3H). Therefore, we hypothesized that macbecin II may exert a synergistic effect in combination with PBMCs on cancer cells. To test this hypothesis, we co-cultured PBMCs (1:10 and 1:20 ratio) with DCIS.com cells in the presence of low doses (0.1 and 0.5 μM) of macbecin II. We found that macbecin II synergistically reduced cancer cell viability in the presence of PBMCs as calculated by the combination index method using CompuSyn (Chou and Talalay, 1984) (Fig. 3D,E). DCIS patients are usually treated with a combination of mastectomy and radiation, but it is still unclear which lesions will eventually progress to the invasive form (Buchheit et al, 2024). We hypothesized that the IL-2-ep13nsEV vaccine prepared from DCIS lesions confers immunity against the recurrent form of cancer. To examine this possibility, we used a system of syngeneic cell lines derived from MCF10A; a DCIS.com cell line derived from MCF10A represented DCIS, while the progressive MCF10CA1a represented the invasive form (Worsham et al, 2006). Therefore, the PBMCs educated with IL-2-ep13nsEV derived from DCIS.com lysate is expected to induce cell death in MCF10CA1a cells. To test this hypothesis, we co-

cultured MCF10CA1a cells with PBMCs and found that PBMCs educated with IL-2-ep13nsEVDCIS.com induced cell death in MCF10CA1a cells (Fig. 3F), while no significant cytotoxic effect was found on MDA-MB-231 cells, which belong to a different haplotype (Fig. EV3I). To test the effect of macbecin II in this system, we co-cultured MCF10CA1a cells with macbecin II (0.1 and 0.5 μM) and PBMCs (1:10 and 1:20 ratio) to evaluate cytotoxicity and found that the combination treatment induced tumor cell death in a synergistic manner (Fig. 3G,H). We also confirmed these results in another breast cancer cell line, E0771. We first isolated T cells from the spleen of an E0771-bearing mouse and activated them with IL-2-ep13nsEV isolated from BMDCs pulsed with a combination of E0771 lysate, IL2-MFG8, C-diGMP, and LPS. The activated T cells were then co-cultured with E0771 cells in combination with macbecin II. We found that the combination of macbecin II and T cells synergistically suppressed cell viability of E0771 cells (Fig. EV3J,K). Our results indicated that macbecin II did not affect the activity or viability of PBMCs (Fig. EV3L,M), suggesting that macbecin II primarily targets cancer cells to induce cell death through enhanced MHC-I expression. These results reveal that macbecin II, in combination with IL-2-ep13nsEV-educated immune cells potentiates cancer cell death and remains effective against progressive breast cancer.

## Macbecin II potentiates anticancer efficacy of IL2-ep13nsEV vaccine in vivo

To examine the efficacy of macbecin II in combination with effector cells of the immune system in vivo, we used an E0771 breast cancer model. We first generated IL-2-ep13nsEV from the BMDCs of syngeneic mice (Fig. 4A). Then, to mimic the clinical progression of DCIS to invasive ductal carcinoma (IDC), we intraductally implanted the E0771 cells in syngeneic mice and treated the animals with macbecin II (2 mg/kg) and IL-2-ep13nsEV (Fig. 4B). We found that IL-2-ep13nsEV alone and in combination with macbecin II significantly reduced tumor growth (Fig. 4C), tumor weight (Fig. 4D), and lung metastasis (Fig. 4E). The combination treatment enhanced total intra-tumoral CD8 (Figs. 4F and EV4A), CD4 (Fig. 4G), as well as activated CD4 (Fig. 4H) and CD8 (Fig. 4I)

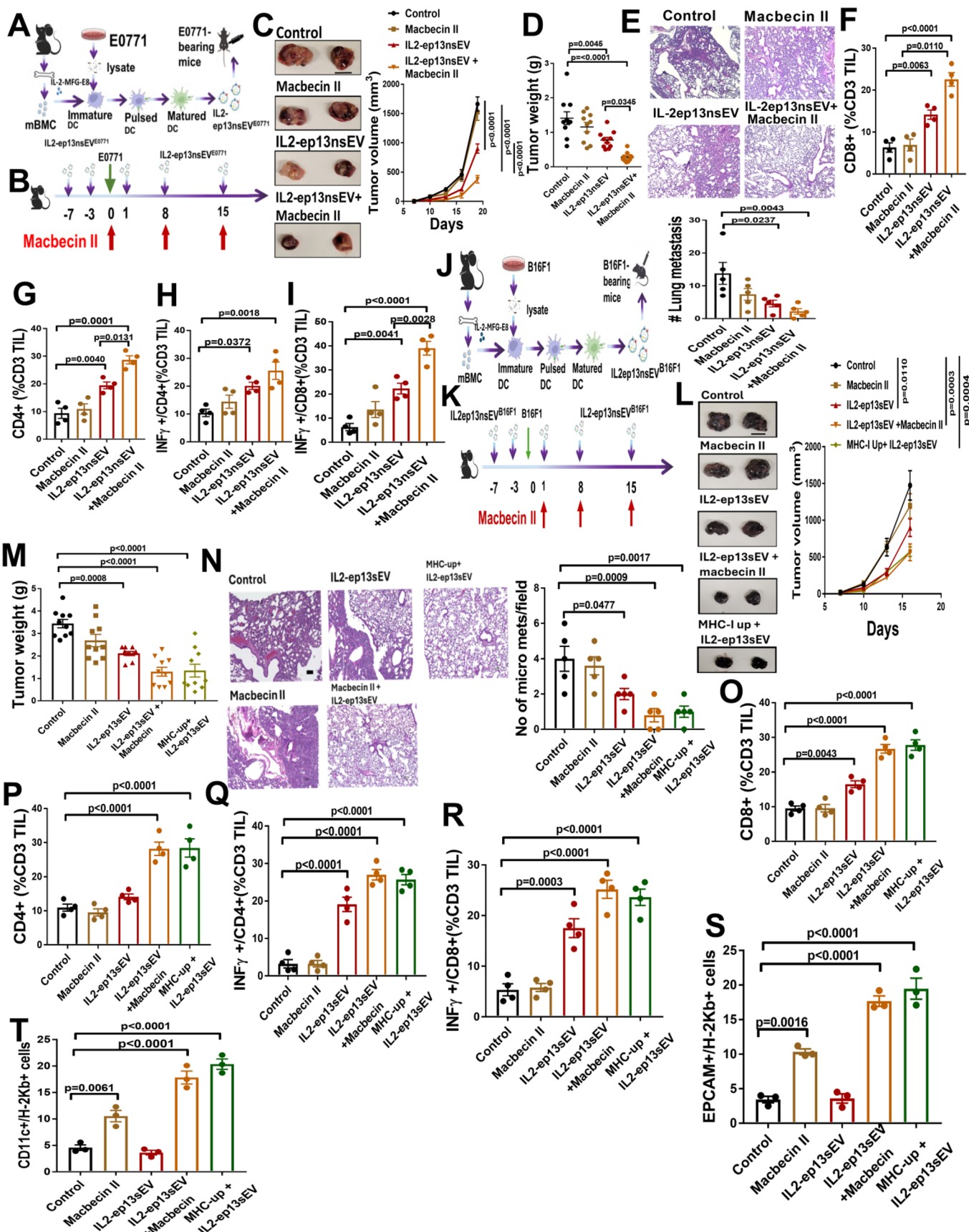

◀

**Figure 4.  Macbecin II potentiates anticancer efficacy of IL2-ep13nsEV vaccine in vivo.**

(A) Schematic of the experiment. (B) E0771 cells (200 K cells) were implanted in syngeneic mice by intraductal injection. Animals were injected with the vaccine alone or in combination with macbecin II at the indicated time and dose. (C) Representative tumor pictures were taken at the endpoint (left). Tumor growth was monitored and compared by two-way ANOVA ($n = 10$/group, biological replicates). Scale bar, 1 cm. (D) Tumor weight was measured at the endpoint and compared by one-way ANOVA with a Tukey post-hoc test ($n = 10$/group, biological replicates). (E) Representative HE images showing lung micro-metastases from each group (up). Lung metastases were measured (down) and compared by one-way ANOVA ($n = 5$/group, biological replicates). Scale bar, 100 μm. (F, G) CD8 (F) and CD4 (G) TILs among the CD3+ cells were measured in the dissociated tumor by FACS and compared by one-way ANOVA with a Tukey post-hoc test ($n = 4$/group, biological replicates). (H, I) IFN-γ positive CD4 (H) and CD8 (I) TILs were measured in the dissociated tumor by FACS and compared by one-way ANOVA with a Tukey post-hoc test ($n = 4$/group, biological replicates). (J, K) Schematic of the experiment. B16-F1 cells ($2 \times 10^5$ cells) with or without ectopic MHC-I expression were implanted in syngeneic mice by subcutaneous injection. Animals were treated with the IL2-ep13nsEV alone or in combination with macbecin II at the indicated time and dose. (L) Representative tumor images were taken at the endpoint (left). Tumor growth was monitored, and the result was analyzed by the two-way ANOVA ($n = 10$/group, biological replicates) test. Scale bar, 1 cm. (M) Tumor weight was measured at the endpoint and the result was analyzed by the one-way ANOVA with a Tukey post-hoc test ($n = 10$/group, biological replicates). (N) Representative HE images showing lung micro-metastases from each group (left). Lung metastases were measured (right) and the result was analyzed by the one-way ANOVA ($n = 5$/group, biological replicates) test. Scale bar, 100 μm. (O, P) CD8 (O) and CD4 (P) TILs among the CD3+ cells were measured in the dissociated tumor by FACS, and the result was analyzed by the one-way ANOVA with a Tukey post-hoc test ($n = 4$/group, biological replicates). (Q, R) IFN-γ positive CD4 (Q) and CD8 (R) TILs were measured in the dissociated tumor by FACS, and the result was analyzed by the one-way ANOVA with a Tukey post-hoc test ($n = 4$/group, biological replicates). (S, T) MHC-I expression was examined in tumor (S) and DC (T) cells in dissociated tumor by FACS and the result was analyzed by the one-way ANOVA with a Tukey post-hoc test ($n = 3$/group, biological replicates). Data are presented as mean $+/-$ SEM. Source data are available online for this figure.

lymphocytes. The combination treatment significantly increased the number of cells that expressed Granzyme B (Fig. EV4B) and LAMP1 (Fig. EV4C) in animals treated with combination of macbecin II and IL2-ep13sEV. We also examined MHC-I protein expression and found that the combination of IL-2-ep13nsEV and macbecin II significantly promoted MHC-I expression on both dendritic and cancer cells (Fig. EV4D,E). The treatment did not affect animal weight (Fig. EV4F) or serum AST activity (Fig. EV4G), indicating limited toxic effects of the treatment regime. To further examine whether the enhanced efficacy of macbecin II in combination with IL-2-ep13nsEV is associated with increased MHC-I expression, we ectopically expressed mouse MHC-I in B16-Ova and E0771 cell lines. Subsequently, we isolated IL-2-ep13nsEV from BMDC pulsed with tumor cell lysate, and they were co-cultured with CD8 T cells that were isolated from the spleens of syngeneic mice. The T cells were then co-cultured with the cancer cells, and cell viability was assessed. Our results demonstrate that MHC-I overexpression reduced cell viability, with an effect comparable to macbecin II treatment (Fig. EV4H,I) when co-cultured in the presence of T cells. We further validated the efficacy of IL-2-ep13nsEV in sensitizing cell death in combination with MHC-I upregulation using an MHC-I-deficient (Peter et al, 2001; Seliger et al, 2001) B16-F1 melanoma model (Fig. EV4J). The ectopic expression of MHC-I (Fig. EV4J) did not affect the cell viability (Fig. EV4K). IL-2-ep13nsEVB16F1 was generated from BMDCs that were pulsed with B16-F1 lysate (Fig. 4J). B16-F1 cells, with or without MHC-I upregulation, were then implanted in syngeneic mice and they were treated with macbecin II (2 mg/kg) and IL-2-ep13nsEVB16F1 (Fig. 4K). Similar to the E0771 breast cancer model, we found that IL-2-ep13nsEVB16F1 alone and in combination with macbecin II significantly reduced tumor growth (Fig. 4L), tumor weight (Fig. 4M), and lung metastasis (Fig. 4N). The combination of IL-2-ep13nsEVB16F1 and macbecin II enhanced the total intra-tumoral CD8+ (Fig. 4O) and CD4+ (Fig. 4P) populations, as well as the activated CD4+ (Fig. 4Q) and CD8+ (Fig. 4R) lymphocytes. The MHC-I upregulation phenocopied the effect of macbecin II treatment. Importantly, the treatment did not affect animal weight (Fig. EV4L) or serum AST activity (Fig. EV4M), indicating minimum toxic effects of the combination regimen. We also examined MHC-I protein expression on cancer

cells and tumor-infiltrating dendritic cells. The combination of IL-2-ep13nsEV B16F1 and macbecin II significantly upregulated MHC-I on tumor (Fig. 4S) and dendritic cells (Fig. 4T). Thus, the MHC-I upregulation mimicked the effects of macbecin II in reducing tumor growth, mitigating lung metastasis, and enhancing CD4+/CD8+ lymphocyte infiltration.

## Macbecin II potentiates anticancer efficacy of anti-PD-1 immune checkpoint blockade in breast cancer

Furthermore, because anti-PD-1 immunotherapy reinvigorates T cells and potentiates breast cancer cell death (Li et al, 2020), we examined whether macbecin II increases the efficacy of anti-PD-1 immune checkpoint blockade. The CD8 T cells were isolated from the spleens of syngenetic mice implanted with E0771 cells. The cells were activated and expanded with CD3/CD28 beads and co-cultured with the cancer cells in combination with anti-PD-1 immune checkpoint blockade. Our results indicated that macbecin II showed a synergistic effect in combination with anti-PD-1 in E0771 cells in vitro (Fig. EV5A,B). To examine the effect of anti-PD-1 and macbecin II in vivo, we implanted the E0771 cells in the mammary ducts of syngeneic mice and treated them with anti-PD-1 alone or in combination with macbecin II (Fig. 5A). The combination of anti-PD-1 and macbecin II was found to significantly reduce tumor growth (Fig. 5B), tumor weight (Fig. 5C), and lung metastasis (Fig. 5D). In addition, the combination of anti-PD-1 and macbecin II promoted the proliferation of CD4 and CD8 T cells (Fig. 4E) and enhanced MHC-I protein expression (Fig. 4F). However, it did not affect animal weight (Fig. EV5C) or serum AST levels (Fig. EV5D). To further examine the role of macbecin II in potentiating the effect of anti-PD-1 immune checkpoint blockade in the endogenously formed tumors, we used genetically modified B6.FVB-Tg(MMTV-PyVT)634Mul/LellJ mouse which sponta-neously forms the mammary tumors. The animals were treated with macbecin II alone or in combination with anti-PD-1 immune checkpoint blockade when the tumors became palpable. We found that macbecin II potentiated the effect of anti-PD-1 immune checkpoint blockade in reducing the tumor growth (Fig. 5G,H) without affecting the body weight (Fig. 5I). Together, our results indicate that macbecin II significantly potentiates the efficacy of IL-

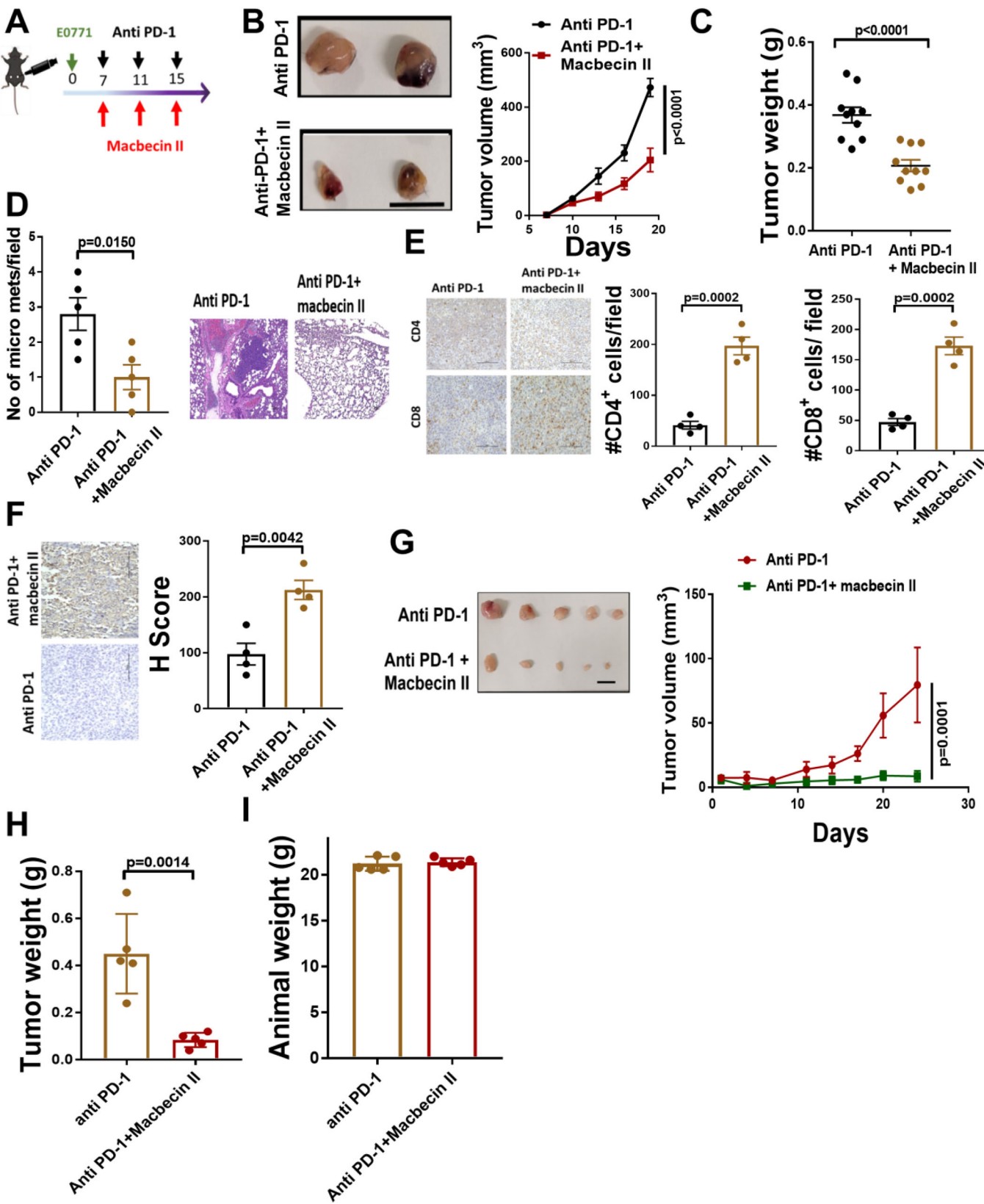

◀ **Figure 5. Macbecin II potentiates anticancer efficacy of anti-PD-1 immune checkpoint blockade in breast cancer.**

(A) Schematic of the experiment. E0771 cells (200 K cells) were implanted in syngeneic mice by intraductal injection. Animals were injected with anti-PD-1 alone or in combination with macbecin II at the indicated time and dose. (B) Representative tumor pictures were taken at the endpoint (left). Tumor growth was monitored and compared by two-way ANOVA ($n = 10$/group, biological replicates). Scale bar, 1 cm. (C) Tumor weight was measured at the endpoint and compared by one-way ANOVA ($n = 10$/group, biological replicates) with a Tukey post-hoc test. (D) Representative HE images showing lung micro-metastases from each group (right). The lung micro-metastases were quantified in random fields from the lung tissue of each mouse (left) ($n = 5$/group, biological replicates). Statistical significance between the groups was determined by an unpaired two-tailed Student's t-test. Scale bar, 100 µm. (E) CD8/CD4 expression was examined through immunohistochemistry (left) and quantified (right) in the two groups ($n = 4$/group, biological replicates). (F) MHC-I expression was examined through immunohistochemistry (left) and quantified (right) in the two groups ($n = 4$/group, biological replicates). The staining intensity was compared by an unpaired two-tailed Student's t-test. Scale bar, 100 µm. (G) Representative tumor images were taken at the endpoint (left). Tumor growth was monitored, and the result was analyzed by the two-way ANOVA test ($n = 5$/group, biological replicates). Scale bar, 1 cm. (H) Average tumor weight was measured at the endpoint and the result was analyzed by the unpaired two-tailed Student's t-test ($n = 5$/group, biological replicates). (I) Body weight of mice was measured at the end-point ($n = 5$/group, biological replicates). Data are represented as mean +/− SEM. Source data are available online for this figure.

2-ep13nsEV and anti-PD-1 immune checkpoint blockade by promoting T cell activity and MHC-I expression in breast cancer.

## Macbecin II upregulates MHC-I through inhibition of lysosomal degradation

To investigate how macbecin II upregulates MHC-I expression, we first examined MHC-I mRNA in DCIS.com and MCF10CA1a cells following treatment with macbecin II and found that there was no change in mRNA expression (Fig. 6A,B). Similar outcomes were confirmed in E0771 and 4T1 breast cancer cells (Fig. EV6A,B). We then examined p-EIF2 alpha expression, which regulates translation, and noted that macbecin II did not alter the protein expression of p-EIF2 alpha (Fig. EV6C). These results indicate that macbecin II is likely to upregulate MHC-I expression post-translationally. To test this possibility, we treated DCIS.com cells with macbecin II in the presence of cycloheximide, which inhibits translation, and examined MHC-I protein stability. We found that MHC-I protein was degraded over time, while this decrease was significantly suppressed by macbecin II (Fig. 6C). Protein degradation is known to be mediated by proteasomal or lysosomal degradation pathways (Sakanyan et al, 2023). To examine how macbecin II stabilizes MHC-I, we treated the DCIS.com cells with the proteasomal inhibitor MG-132 (Zhu et al, 2023) or the lysosomal degradation inhibitor bafilomycin (Huang et al, 2024). Our results showed that the protein degradation mediated by MG-132 treatment was rescued by the macbecin II (Fig. 6D) treatment, while the lysosomal degradation inhibitor bafilomycin suppressed the MHC-I degradation (Fig. 6E), indicating that macbecin II inhibits lysosomal degradation to prevent MHC-I downregulation. To confirm this result, we treated the DCIS.com cells with C381, which is known to promote lysosomal acidification (Vest et al, 2022), in the presence of macbecin II and found that decreased MHC-I expression was significantly suppressed by macbecin II (Fig. 6F,G). We also examined the expression of MHC-II and found that macbecin II did not regulate the RNA expression of MHC-II (Fig. EV6D,E), but it promoted the stability of MHC-II protein (Fig. EV6F). Similar to MHC-I, we found that inhibiting proteasomal degradation did not affect the stability of MHC-II (Fig. EV6G), while the lysosomal degradation inhibitor bafilomycin stabilized MHC-II expression (Fig. EV6H). In addition, the total expression of MHC-II protein was rescued when treated with C381 and macbecin II (Fig. EV6I). We examined PD-L1 expression on tumor cells following macbecin II treatment and found no

significant change in PD-L1 expression. This demonstrates that macbecin II does not have widespread effects on other cell surface receptors (Fig. EV6J,K). Together, our results indicate that macbecin II promotes MHC-I and II expression by inhibiting lysosomal degradation.

## Discussion

In this study, we screened a natural compound library and identified macbecin II as a reagent to boost the expression of MHC-I on early and late-stage breast cancer cells to overcome their resistance to immunotherapies. Using a breast cancer progression model, we have shown that macbecin II significantly synergized the efficacies of tumor vaccines and immune checkpoint blockade by promoting immune cell infiltration and inducing antigen-dependent cancer cell death.

Cancer cells often suppress dendritic cell (DC) maturation and expression of cell surface MHC-II, leading to compromised antigen presentation (Gervais et al, 2005). Moreover, MHC-I is heterogeneously expressed in breast cancer and the loss of tumor-specific MHC-I was found to confer resistance to immunotherapy (Lee et al, 2020). Thus, in addition to low mutation burden, inefficient antigen presentation and low immune cell infiltration are considered to contribute to reduce the immunotherapy response rate among breast cancer patients to as low as 12% (Swoboda and Nanda, 2018). Presently, DCIS patients are treated with lumpectomy or mastectomy as it is unclear which lesions will progress to invasive ductal carcinoma (Bae et al, 2024). However, these are considered overtreatment, and therefore, less aggressive and personalized approaches such as immunotherapy or vaccines may be more appropriate. We recently developed the IL2-ep13nsEV vaccine using sEVs from DCs by uniquely expressing bioactive IL2 and loading it with personalized tumor antigens, followed by maturation with LPS and STING agonist to express co-stimulatory factors to increase its interaction with T cells. We have shown that IL2-ep13nsEV can present antigens and activate T cells independent of APCs (Wu et al, 2023a). In this study, we investigated the effect of macbecin II on IL2-ep13nsEV in our breast cancer progression model with intraductal tumor implantation. Our results showed that IL2-ep13nsEV promoted CD8 T cell activity through MHC-I dependent antigen presentation mediated by macbecin II, followed by suppression of tumor growth. Therefore, our results indicate a promising utility of the combination of

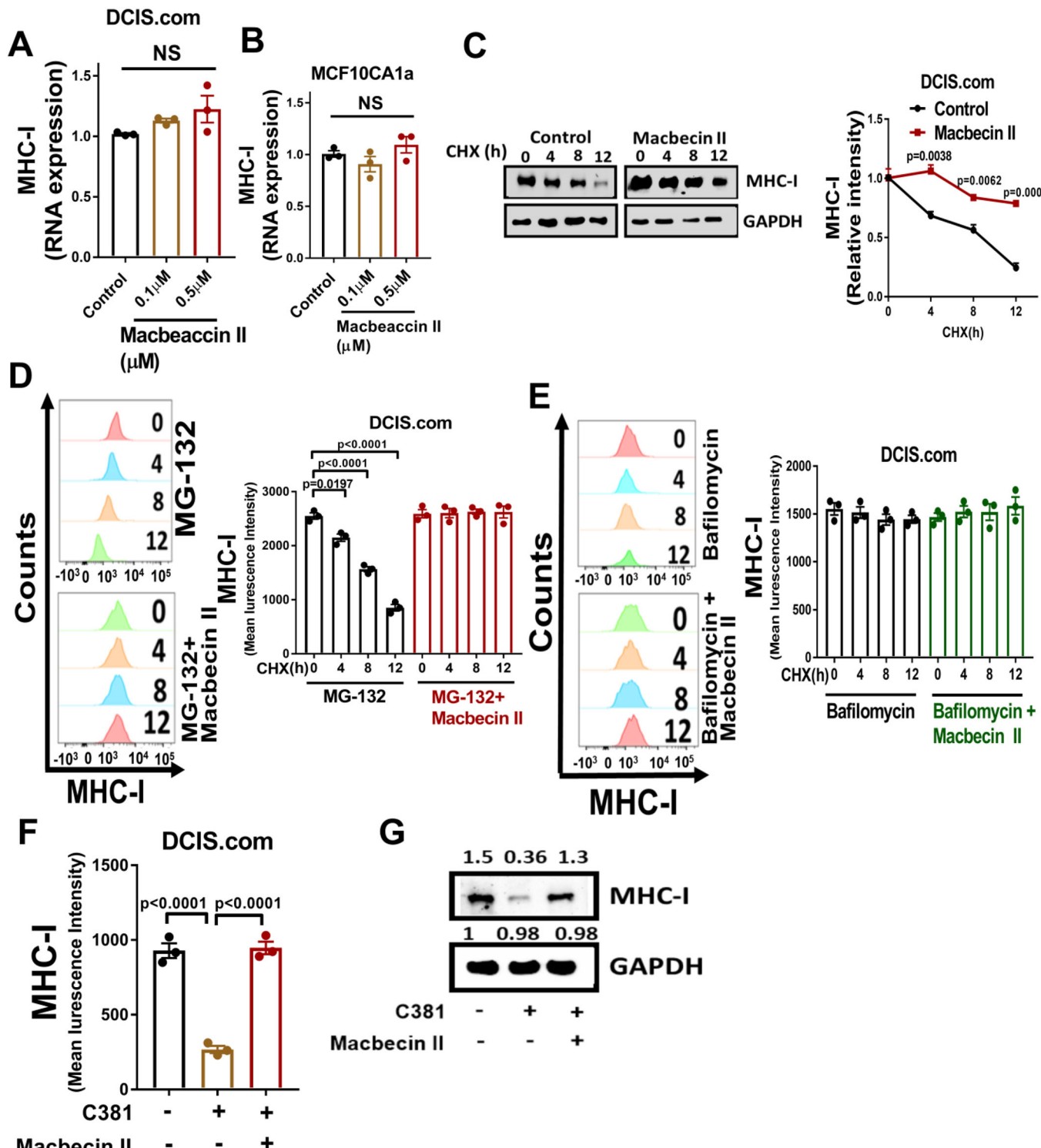

macbecin II and personalized tumor vaccine for the treatment of DCIS. Furthermore, we tested the efficacy of macbecin II with immune checkpoint blockade. The tumor-infiltrated CD8 T cells are known to upregulate PD-1 expression on the cell surface, which interacts with PD-L1 on the cancer cells (Elfoly et al, 2024). This PD-1-PD-L1 interaction is reported to exhaust the CD8 T cells, resulting in immune evasion of cancer cells (Yilmaz et al, 2023). To overcome this limitation, we employed the combination of macbecin II and anti-PD-1 immune checkpoint blockade to reinvigorate the exhausted T cells and enhance antigen presentation. We found that the combination treatment significantly reduced tumor burden and mitigated the metastatic potential of

**Figure 6. Macbecin II upregulates MHC-I through inhibition of lysosomal degradation.**

(A, B) DCIS.com (A) and MCF10CA1a (B) cells were seeded in a 24-well plate and treated with the indicated dose of macbecin II for 48 h. Total RNA was isolated, and cDNA synthesis was performed with the iScript cDNA Synthesis Kit followed by SYBR green-based real-time PCR. GAPDH was used as an internal control. Statistical significance was determined by one-way ANOVA with a Tukey post-hoc test ($n = 3$/group, biological replicates). (C) DCIS.com cells were treated with cycloheximide (50 μg/mL) and macbecin II (0.5 μM) at the indicated time and MHC-I expression was examined by western blot ($n = 3$/group, biological replicates). (D) DCIS.com cells were treated with vehicle, a combination of MG-132 (10 μM) and macbecin II (0.5 μM), or MG-132 alone in the presence of cycloheximide (50 μg/mL) at the indicated time and MHC-I expression was examined by FACS. Statistical significance was determined by one-way ANOVA with a Tukey post-hoc test ($n = 3$/group, biological replicates). (E) DCIS.com cells were treated with vehicle, a combination of Bafilomycin (100 nM) and macbecin II (0.5 μM), or Bafilomycin alone in the presence of cycloheximide (50 μg/mL) at the indicated time and MHC-I expression was examined by FACS ($n = 3$/group, biological replicates). (F) DCIS.com cells were treated with vehicle, a combination of macbecin II (0.5 μM) and C381 (30 μM), or C381 alone in the presence of cycloheximide (50 μg/mL) for 12 h and MHC-I expression was examined by FACS. Statistical significance was determined by one-way ANOVA with a Tukey post-hoc test ($n = 3$/group, biological replicates). (G) DCIS.com cells were seeded in a 24-well plate and cultured in the presence of cycloheximide (50 μg/mL). Cells were treated with the vehicle, a combination of C381 (30 μM) and macbecin II (0.5 μM), or C381 alone for 12 h and MHC-I expression was examined by western blot. Data are represented as mean $+/-$ SEM. Source data are available online for this figure.

cancer cells, along with augmenting immune cell infiltration. This suggests that macbecin II can be utilized to augment the efficacy of checkpoint inhibitors for therapy-resistant breast cancer.

Macbecin II was first isolated as an anti-microbial agent from the Nocardia species and later shown to exert antitumor activity against murine leukemia and prostate cancer in preclinical models at a dose of 10–60 mg/kg (Martin et al, 2008; Muroi et al, 1980). In the present research, we showed that a low dose of macbecin II (2 mg/kg) that does not directly affect tumor growth synergistically augmented the efficacy of tumor vaccine and immune checkpoint blockade (Fig. 4). It should be noted that macbecin II was previously found to inhibit HSP90 by binding the ATP-binding site at the N-terminal (Martin et al, 2008). In many cancers, higher HSP90 expression was shown to be associated with larger tumor size, increased lymph node metastasis, and poorer overall survival (Pick et al, 2007), and several small molecule HSP90 inhibitors were shown to display anticancer activity (Neckers et al, 2007; Tsutsumi et al, 2009). Macbecin II was also shown to effectively inhibit HSP90 client proteins at a dose of 10 μM in cancer cells (Martin et al, 2008). On the other hand, we showed that macbecin II upregulated MHC-I at a much lower dose of 0.1–0.5 μM. Therefore, the mechanism of upregulation of MHC-I by macbecin II is considered independent of its anti-HSP90 activity. In fact, we have shown that macbecin II stabilized MHC-I protein by suppressing lysosomal degradation but not proteasomal degradation which is known to be HSP90 dependent (Li et al, 2017). In this context, it is worth mentioning that suppression of MHC-I by preventing proteasomal degradation was also shown to augment the efficacy of immune checkpoint blockade (Mbofung et al, 2017; Rahmy et al, 2022). Therefore, dual inhibition of lysosomal and proteasomal degradation of MHC-I may be an effective clinical strategy for the treatment of cancers that are resistant to checkpoint inhibitors. In addition, macbecin II may possibly modify the lysosomal degradation through the following mechanisms: (i) Inhibiting the translocation of MHC-I in lysosomes. (ii) Increasing the lysosomal acidification. (iii) Macbecin II may inhibit RAB7b and /or promote RAB11 to increase the membrane MHC-I localization.

In summary, our work identified macbecin II as a promoter of MHC-I expression, which helps induce antigen-dependent cell death in early and late-stage breast cancer. We also showed that macbecin II boosted T cell infiltration and enhanced the efficacy of tumor vaccines as well as anti-PD-1 immunotherapy by inhibiting the lysosomal degradation of MHC-I. Therefore, our findings

provide valuable information on a mechanism to limit early-stage breast cancer progression and have the potential to affect the clinical management of both early and late-stage breast cancer.

## Methods

**Reagents and tools table**

| Reagent/Resource | Reference or Source | Identifier or Catalog Number |
|---|---|---|
| **Experimental models** | | |
| E0771 | ATCC | |
| 4T1 | ATCC | |
| MDA-MB-231 | ATCC | |
| MCF10CA1a | ATCC | |
| B16-Ova | ATCC | |
| B16F1 | ATCC | |
| C57BL/6 mouse | The Jackson Laboratory | |
| B6.FVB-Tg(MMTV-PyVT)634Mul/LellJ | The Jackson Laboratory | |
| **Recombinant DNA** | | |
| MHC I overexpression plasmid | Addgene | 111623 |
| pLVX-puro-cOVA | Addgene | 135073 |
| **Antibodies** | | |
| GAPDH antibody | CST | 5174S |
| Recombinant Anti-MHC class I+HLA A+HLA B antibody | Abcam | ab134189 |
| Goat Anti-Rabbit IgG H&L (Alexa Fluor) | Abcam | ab150077 |
| MHC II antibody | BioLegend | 307602 |
| Alexa fluor 488 goat anti-mouse antibody | Invitrogen | A11029 |
| H-2Kb- ova antibody | BioLegend | 141603 |
| InVivoMAb anti-mouse PD-1 | Bioxcell | CD279 |

| Reagent/Resource | Reference or Source | Identifier or Catalog Number |
|---|---|---|
| CD69 Antibody | Biolegend | 104507 |
| InVivoMAb anti-mouse MHC Class I | Bioxcell | BE0077 |
| PE anti-mouse CD4 antibody | Biolegend | 100407 |
| APC anti-mouse CD8a antibody | Biolegend | 100711 |
| Anti-HLA-DR antibody | Sino Biologicals | 105342-T40 |
| FITC anti-mouse CD3 antibody | Biolegend | 100203 |
| PE/Cyanine7 anti-mouse IFN-γ antibody | Biolegend | 505825 |
| Granzyme B antibody | Abclone | A2557 |
| LAMP1 antibody | Bioss | BS-1970R |
| CD326 antibody | Biolegend | 118213 |
| CD11c antibody | Biolegend | 117309 |
| MHC-I antibody | Invitrogen | 11-5958-80 |
| PD-L1 (D4H1Z) Rabbit mAb | CST | 60475 |
| Phospho-eIF2α antibody | CST | 3398T |
| **Oligonucleotides and other sequence-based reagents** | | |
| Primers are listed in Table EV1 | | |
| **Chemicals, Enzymes and other reagents** | | |
| DMEM | Gibco | 11965092 |
| RPMI 1640 Medium | Gibco | 11875093 |
| FBS | Gibco | 16000044 |
| Penicillin-Streptomycin | Gibco | 15140122 |
| Mycoplasma removal agent (MRA) | MP Biomedical | 3050044 |
| Recombinant human GM-CSF | Peprotech | 212-12 |
| Recombinant Murine GM-CSF | Peprotech | 315-03 |
| Lipopolysaccharide (LPS) Solution | Thermo Fisher Scientific | 00-4976-93 |
| Recombinant Murine IL-2 | Peprotech | 212-12 |
| Recombinant Human IL-2 | Peprotech | 200-02 |
| Laemmli sample buffer | Bio-Rad | 161-0737 |
| CellTiter 96® AQueous One Solution Cell Proliferation Assay (MTS) | Promega | G3582 |
| RBC Lysis Buffer (10X) | BioLegend | 420301 |
| crystal violet | Fisher | S25275B |
| SuperSignal™ West Pico PLUS Chemiluminescent Substrate | Thermo Scientific | 34580 |
| Intracellular Staining Permeabilization Wash Buffer | BioLegend | 421002 |

| Reagent/Resource | Reference or Source | Identifier or Catalog Number |
|---|---|---|
| Ova peptide | GenScript | RP10611 |
| MG132 | Adooq Bioscience | A11043 |
| Bafilomycin | Adooq Bioscience | A11961 |
| PE-Labeled Mouse H-2Kb&B2M&OVA (SIINFEKL) Tetramer Protein | ACROBiosystems | H2A-MP2H7-25ug |
| **Commercial assays** | | |
| CD3/CD28 activation kit | ThermoFisher | 10971 |
| T Cell Activation/ Expansion Kit, mouse | Miltenyi Biotec | 130-093-627 |
| RNA isolation kit | Zymo | R2072 |
| CD8a+ T Cell Isolation Kit, mouse | Miltenyi | 130-095-236 |
| iScript CDNA synthesis kit | Bio-Rad | 1708890 |
| ImmPACT® DAB EqV Substrate Kit, Peroxidase (HRP) | Vector Labs | SK-4103 |
| Quick Start™ Bradford Protein Assay | Bio-Rad | 5000201 |
| AST assay kit | MilliporeSigma | MAK055-1KT |
| Zombie Aqua Fixable Viability Kit | BioLegend | 423101 |
| Mouse TNF-alpha ELISA Kit | RayBiotech | ELM-TNFa-1 |
| **Software** | | |
| GraphPad Prism (V 7.0) | http://www.graphpad.com/ | |
| Compysyn | https://compusyn.software.informer.com/ | |
| ImajeJ | https://ij.imjoy.io/ | |

## Cell lines and culture methods

Human breast cancer cell lines (DCIS.com, MCF10CA1a, and MDA-MB-231) were purchased from ATCC and cultured in RPMI 1640 medium (Invitrogen) with 10% FBS (fetal bovine serum). Mouse breast cancer cell lines (E0771 and 4T1) were obtained from ATCC and cultured in RPMI 1640 medium (Invitrogen) with 10% FBS. The mouse melanoma cell line B16-OVA was purchased from Millipore Sigma and maintained in RPMI 1640 supplemented with 10% FBS, 1X nonessential amino acids, 10 mM HEPES, 54 μM β-mercaptoethanol, and geneticin (1 mg/ml). Mouse bone marrow DCs (BMDCs) were isolated and maintained as described previously (Näslund et al, 2013; Wang et al, 2016). Briefly, the bone marrow cells were collected by flushing the femur and tibia bones of 6- to 8-week-old C57BL/6J mice with ice-cold RPMI 1640 medium. After collection, RBCs were lysed, and the cells were cultured in RPMI 1640 with 10% FBS and GM-CSF (20 ng/ml) for 6 days to separate the non-adherent and adherent cells. The loosely adherent and floating cells were collected and educated with the

tumor lysate for 24 h, followed by treatment with LPS (100 ng/ml) and C-diGMP (200 μM). Human primary PBMCs were used to generate the DCs, as mentioned previously (Chometon et al, 2020). Briefly, human PBMCs were cultured in RPMI 1640 with human IL-4 (300 UI/ml) and recombinant human GM-CSF (500 UI/ml). The differentiated DCs were then collected and pulsed with tumor lysate (100 μg/ml) for 24 h, followed by treating them with LPS (100 ng/ml) and C-diGMP (200 μM) for DC stimulation (Karaolis et al, 2007). The cells were authenticated and tested for mycoplasma contamination. The complete list of used reagents along with the identifiers are provided as Reagents Table.

## Generation of p13nsEV tumor vaccine

p13nsEVs were generated as mentioned previously (Wu et al, 2023a). A plasmid encoding the IL2-MFG-E8 fusion protein (System Biosciences) was used to engineer IL2 expression on the surface of p13nsEVs. IL-2 lentivirus was produced by transfecting 800 ng of transfer plasmid, 600 ng of viral packaging plasmid pPAX2, and 200 ng of envelope plasmid pMD2.G into human embryonic kidney (HEK293T) cells in a 10 cm dish. The monocytes were differentiated and pulsed with tumor lysate. To isolate the p13nsEVs, the cells were grown in exosome-depleted media for 48 h, and conditioned medium (CM) was collected for p13nsEV isolation by sequential ultracentrifugation as described before (Théry et al, 2006). In brief, the CM was centrifuged at $300 \times g$ for 10 min to remove the cells after which the cell debris was removed by centrifugation at $2000 \times g$ for 20 min. Then, the p13nsEVs were removed by centrifuging the supernatant at $16,500 \times g$ for 20 min. The supernatant was passed through a 0.2-μm filter (Sarstedt) to remove particles with a diameter larger than 200 nm. The p13nsEVs were then pelleted by ultracentrifugation at $120,000 \times g$ for 70 min.

## Analysis of extracellular vesicles by transmission electron microscopy

The extracellular vesicles were visualized using negative staining through electron microscopy. The vesicles were purified and placed onto discharged 200-mesh copper EM grids, followed by fixation with 2% paraformaldehyde (PFA). A 1% uranyl acetate solution was used to stain the extracellular vesicles on the EM grids, which were then imaged using an FEI Tecnai BioTwin Transmission Electron Microscope.

## Animal experiments

The animal experimental protocols were approved by the Institutional Animal Care and Use Committee at Wake Forest Health Sciences. All animals were obtained from the Jackson Laboratory. The intraductal injection procedure was performed as described previously (Oliemuller et al, 2022). Briefly, the animals were shaved to remove hair and 0.2 M cells were injected under a microscope using the Hamilton syringe into the 3rd nipple of 6- to 8-week-old female C57BL/6J mice. To investigate the effect of macbecin II and active immunotherapy on the primary tumor, IL2-ep13nsEV was given by intramuscular injection into the tail base of the mice at 50 μg each time on days 3 and 7 before and 1, 8, and 15 days after tumor cell implantation while macbecin II (2 mg/kg) was injected

on day 0, 8, and 15. To investigate the effect of macbecin II and ICI, anti-mouse-PD-1 or IgG (Bio X Cell) was given to the mice at a 20 mg/kg dose by intraperitoneal injection on days 7, 11, 15, and 19 after tumor inoculation. The macbecin II was injected at the same time interval at a 2 mg/kg dose. The growth of tumors in mice was quantified by measuring the length and width of the tumor using an electronic caliper. Tumor volume was calculated by the modified ellipsoidal formula: volume $= 1/2(\text{length} \times \text{width}^2)$. The animals were housed in a 12 h light/12 h dark cycle and fed with standard chow and water ad libitum.

## Natural compound library screening

The Natural compound set V (NCI) was used for the screening to identify a compound that upregulates MHC-I expression. The DCIS.com cells (10,000 cells/well) were seeded in a 96-well plate and treated with the natural compound library (5 μM) for 48 h. The cells were washed with PBS and fixed with ice-cold methanol for 15 min at room temperature. The cells were then incubated with an anti-MHC-I primary antibody (1:500) overnight at 4 °C. After the incubation, the cells were washed with PBS and incubated with a Goat Anti-Rabbit IgG H&L (Alexa Fluor) secondary antibody (1:2000) for 1 h at 37 °C. The cells were then washed with PBS and the readouts were obtained by a fluorescent plate reader (excitation: 480 nm, emission: 520 nm).

## Flow cytometry

The cultured cancer cells were detached with a trypsin-EDTA (0.25%) solution and washed with PBS. The cells were then stained with the Zombie Aqua Fixable Viability Kit (BioLegend) to examine the viable cells. Briefly, the cells were stained with cell surface staining antibodies for 30 min at room temperature. For intracellular staining, the cells were fixed with 4% PFA for 20 min at room temperature. The cells were then treated with a permeabilization buffer (BioLegend) followed by staining with antibodies for 30 min at room temperature. For immunophenotyping of tumors, the tissues were dissociated with 2 mg/ml collagenase and 0.1 mg/ml DNase at 37 °C for 30 min. A single-cell suspension was obtained after filtering through a 70 μm strainer. Then, RBC lysis was performed with RBC lysis buffer for 5 min at 4 °C followed by staining with aqua zombie dye. The cells were first stained with cell surface staining antibodies for 30 min at room temperature and then fixed with 4% PFA in PBS for 20 min. The intracellular staining was done by treating the cells with the permeabilization buffer followed by staining with antibodies at room temperature for 30 min. Cells were then examined by the BD FACSCanto II Flow Cytometer, and the data were analyzed by FlowJo software.

## T-cell activation and cytotoxicity assay

T cells in human PBMCs were activated and expanded after co-culturing with IL2-ep13nsEV (50 μg/ml) for 14 days. Mouse CD8 T cells were isolated from the spleens of syngeneic mice using a CD8 T cell isolation kit. The CD8 T cells were then cultured in the presence of IL2-ep13nsEV generated from BMDCs derived from syngeneic mice for 7 days. RPMI 1640 medium supplemented with 10% FBS was used for PBMC/T cell culture. For the co-culture assay, the cancer cells (500 cells/well) were seeded in a 96-well plate

for 2 h and treated with CD8 T cells or PBMCs in the presence of the designated dose of macbecin II or 40 µg/mL anti-PD-1 immune checkpoint blockade for 48 h. After the incubation, the cells were washed with ice-cold PBS to dislodge the PBMCs and dead cells. The surviving cells were then fixed with ice-cold methanol for 15 min at room temperature and stained with 0.5% crystal violet. The dye was dissolved in 10% acetic acid, and the readouts were measured at 590 nm. The synergy was calculated using the combination index method with CompuSyn. To examine the tumor-infiltrating lymphocytes, the tumor tissues were minced and incubated with the tissue dissociation solution (1 mg/mL collagenase IV, RPMI 1650, 5% FBS, 200 U/mL, DNase) at 370 °C for 20 min. The cells were passed through a 70 µm strainer, and RBC lysis was performed with RBC lysis buffer. The single cells were then incubated with 100 ng/ml PMA and 1 µg/ml ionomycin and 2.5 mg/ml protein transport inhibitor Brefeldin A at 370 °C for 4 h. The dead cells were then stained using the Zombie Aqua Fixable Viability Kit. The cells were then washed with PBS and analyzed or further fixed in 4% PFA in PBS for 15 min for intracellular cytokine staining. For intracellular cytokine staining, cells were treated with permeabilization buffer followed by staining with antibodies for 30 min. The cells were then examined by the BD Canto II Flow Cytometer, and the data were analyzed by FlowJo software.

## AST assay

The AST activity was measured using an AST Activity Assay kit (Sigma-Aldrich) following the manufacturer's instructions. Briefly, serum was isolated from animal blood and used directly for the assay. Glutamate standard solutions were prepared, and the reaction mixture was reconstituted with AST Assay Buffer (80 µL), AST Enzyme Mix (2 µL), AST Developer (8 µL), and AST Substrate (10 µL). Then, 100 µL of the reaction mixture was added to each well. After incubation at 37 °C for 2 to 3 min (Tinitial), the initial absorption measurement was acquired at 450 nm. Measurements were then taken every 5 min until the value of the most active sample exceeded the value of the highest standard. Then, the final reading (Tfinal) was obtained to calculate the enzyme activity (A450). The amount of glutamate generated (B) between Tinitial and Tfinal was calculated by plotting ΔA450 = (A570) final − (A570) initial on the glutamate standard curve. The AST activity was then calculated as: B × sample dilution factor/ [(Tfinal − Tinitial) × sample volume].

## Real-time PCR and western blot

Total RNA was isolated using the Direct-zol RNA miniprep kit according to the manufacturer's instructions. The purity of the isolated RNA was examined using a NanoDrop spectrophotometer, and 200 ng of RNA was used for cDNA synthesis with the cDNA synthesis kit. The synthesized cDNA was then used for SYBR green-based real-time PCR, and the results were analyzed using the relative quantification method. GAPDH was used as an internal control. The list of primer sequences is provided in Table EV1. For western blot, the cell lysates were prepared using RIPA buffer. The proteins in the lysate were quantified using the Bradford protein assay, and 50 µg of protein was loaded onto a 10% acrylamide gel. After separation, the proteins were transferred to the nitrocellulose membrane using a semi-dry method (100 mA, 2 h). The membrane was then blocked

using nonfat skim milk powder (5%) dissolved in PBS, followed by incubation with primary antibodies for MHC-I (1:500), p-eIF2 alpha (1:500), and GAPDH (1:2000) overnight at 4 °C. Blots were then incubated with a secondary antibody (1:2000) and developed with West Pico PLUS Chemiluminescent Substrate and an Amersham Imager, followed by quantification using Image J.

## Immunohistochemistry

The Immunohistochemistry staining was performed as described previously (Deshpande et al, 2022). The stained cells were quantified to calculate the H score manually. Cells stained were categorized as: weak, moderate and strong. Then, H score was calculated with the formula (1 × percentage of weak staining) + (2 × percentage of moderate staining) + (3 × percentage of strong staining). The observer was blended for the group of samples used.

## Statistical analysis

The $P$ value for two groups were calculated with Two-tailed unpaired t tests while one-way ANOVA was used for more than two groups. Data expressed as mean $+/-$ SEM. Significance between each group was represented as $p < 0.05$. The experiments were repeated independently three times, and no values were excluded from the analysis. The animals were randomized before the treatment and blinded before measuring the tumor volume. All the statistical analysis was performed using GraphPad Prism (version 7.0).

## Ethics approval

The animal study was reviewed and approved by the Institutional Animal Care and Use Committee at Wake Forest Health Science.

# Data availability

This study includes no data deposited in external repositories.

The source data of this paper are collected in the following database record: biostudies:S-SCDT-10_1038-S44321-025-00213-7.

# Peer review information

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

## Acknowledgements

This work was supported by grants R01CA205067 from NIH and W81XWH2110075 and W81XWH2210463 from Department of Defense (to KW). This study also utilized the Cancer Center Shared Resources including Cancer Genomics, Tumor Tissue and Pathology, Cell Engineering, Flow Cytometry, and Biostatistics and Bioinformatics that are supported by the Comprehensive Cancer Center of Wake Forest University NCI, National Institutes of Health Grant (P30CA012197).

## Author contributions

**Ravindra Pramod Deshpande**: Conceptualization; Data curation; Formal analysis; Validation; Investigation; Visualization; Methodology; Writing—original draft; Project administration; Writing—review and editing. **Kerui Wu**: Formal analysis. **Shih-Ying Wu**: Formal analysis. **Abhishek Tyagi**: Formal analysis. **Eleanor C Smith**: Formal analysis. **Jee-Won Kim**: Formal analysis. **Kounosuke Watabe**: Conceptualization; Resources; Software; Supervision; Project administration.

Source data underlying figure panels in this paper may have individual authorship assigned. Where available, figure panel/source data authorship is listed in the following database record: biostudies:S-SCDT-10_1038-S44321-025-00213-7.

## Disclosure and competing interests statement

The authors declare no competing interests.

# Expanded View Figures

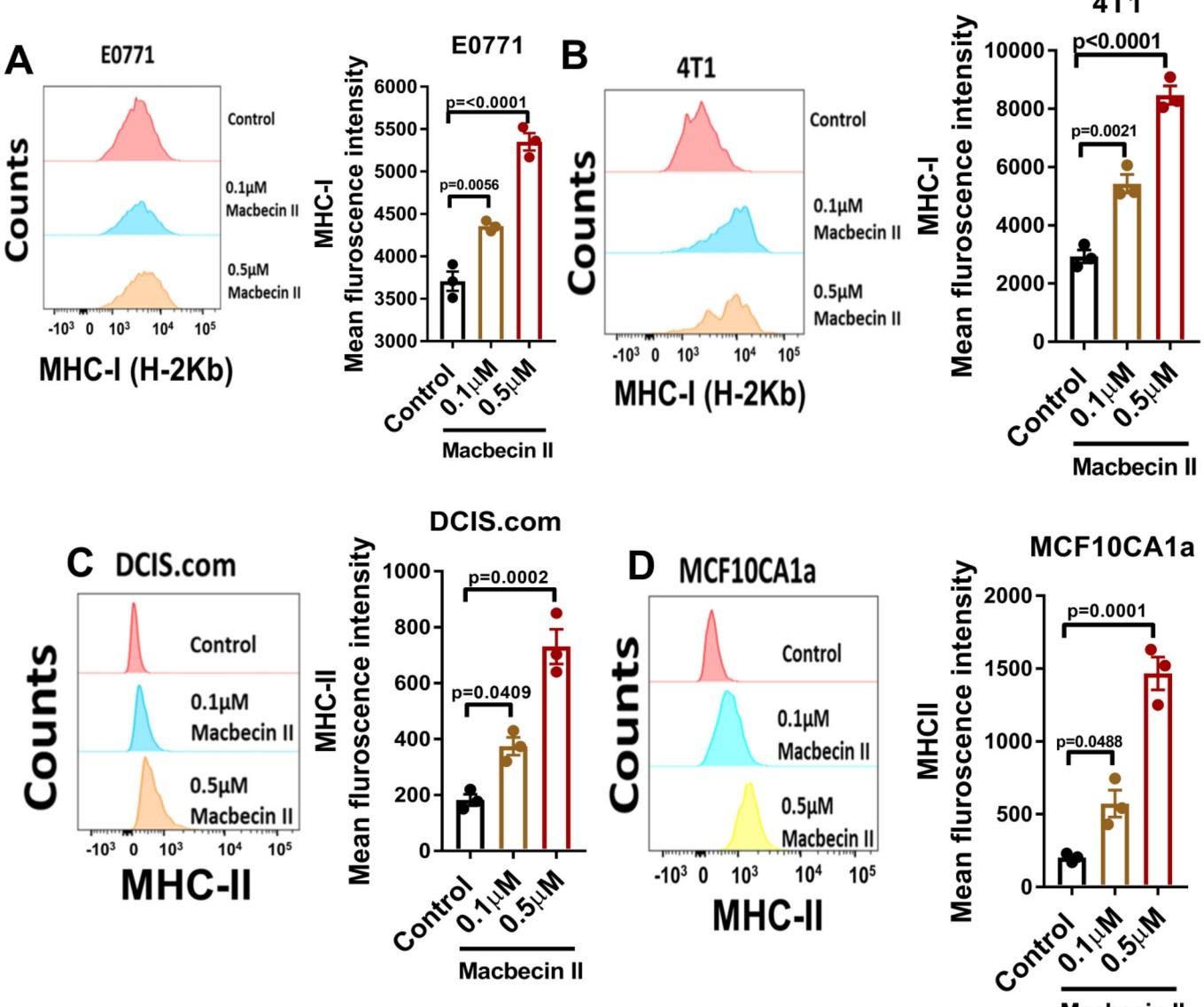

**Figure EV1.  Macbecin II stimulates MHC-I and II expression.**

(A) E0771 cells were seeded in a 24-well plate and treated with the indicated dose of macbecin II for 48 h. Cells were trypsinized and stained with anti-MHC I (H-2Kb) primary and secondary antibody. The readouts were obtained by BD FACS Canto. Mean fluorescence intensity was calculated and compared by one-way ANOVA with a Tukey post-hoc test ($n = 3$/group, biological replicates). (B) 4T1 cells were treated with the indicated dose of macbecin II for 48 h and MHC-I (H-2Kb)expression was examined as described in (A). Mean fluorescence intensity was calculated and compared by one-way ANOVA with a Tukey post-hoc test ($n = 3$/group, biological replicates). (C) DCIS.com cells were treated with macbecin II at the indicated doses for 48 h. The cells were trypsinized and stained with anti-MHC-II primary followed by secondary antibodies. Readouts were obtained using a BD FACS Canto. Mean fluorescence intensity was calculated and analyzed using the one-way ANOVA with a Tukey post-hoc test ($n = 3$/group, biological replicates). (D) MCF10CA1a cells were treated with macbecin II at the indicated doses for 48 h. The cells were processed as described in (C), and the results were analyzed using the one-way ANOVA with a Tukey post-hoc test ($n = 3$/group, biological replicates). Data are represented as mean $+/-$ SEM.

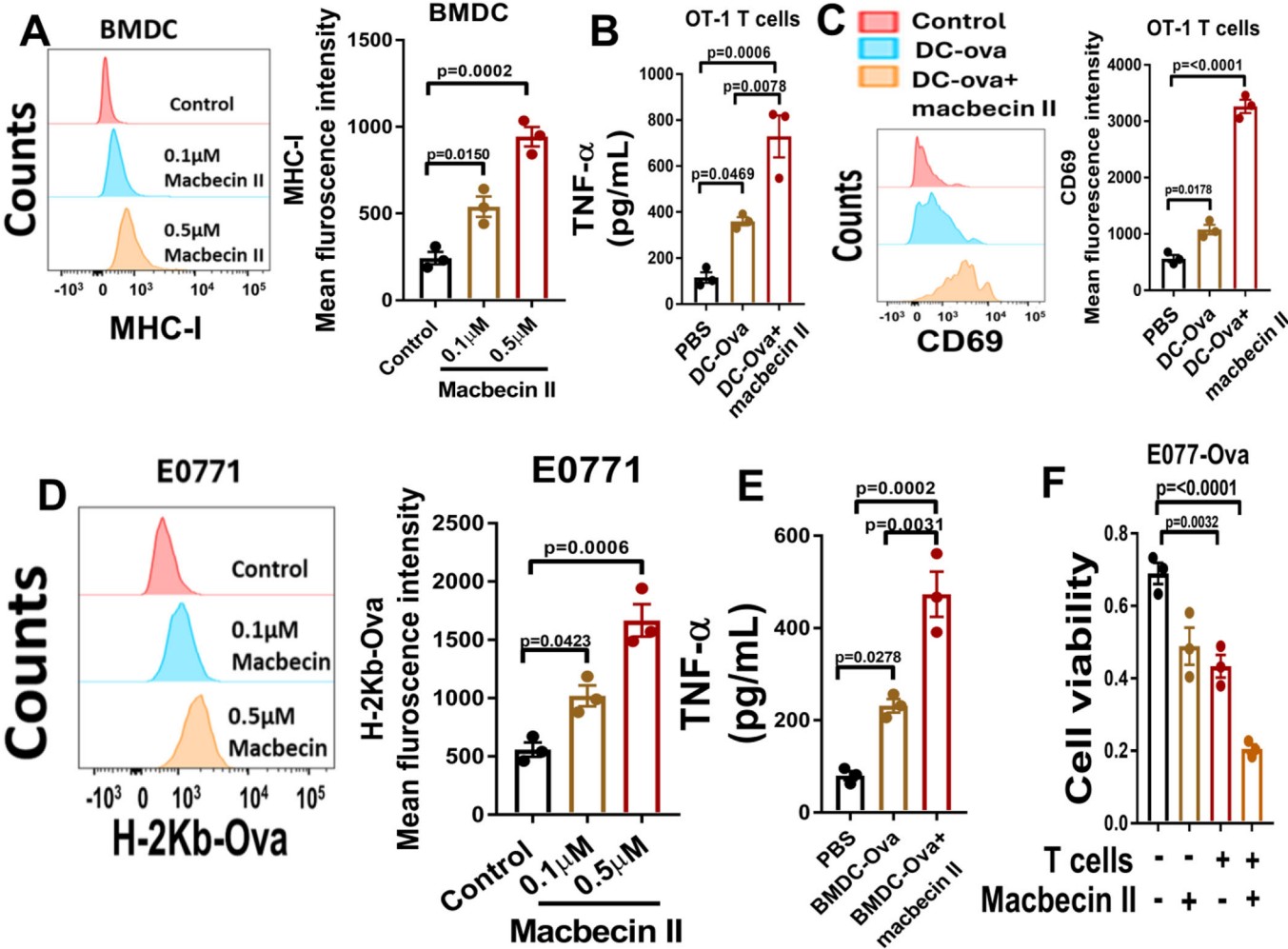

**Figure EV2. Macbecin II promotes antigen presentation and enhances antigen-dependent cancer cell death.**

(A) BMDCs were treated with the indicated dose of macbecin II for 48 h. Post-incubation, cells were stained with an H-2Kb antibody and examined by FACS ($n = 3$/group, biological replicates). Statistical inference was compared by one-way ANOVA with a Tukey post-hoc test. (B) OT-1 cells were co-cultured with B16-Ova cells and the expression of TNF-α in the medium was measured by ELISA. Statistical significance was determined using one-way ANOVA with a Tukey post-hoc test ($n = 3$/group, biological replicates). (C) OT-1 T cells were co-cultured in the presence of DCs that were pulsed with Ova (DC-ova) with or without macbecin II, and then examined for CD69 surface expression by FACS at after three days. Statistical significance was determined using the one-way ANOVA with a Tukey post-hoc test ($n = 3$/group, biological replicates). (D) E0771 cells transfected with the ova plasmid were treated with the indicated doses of macbecin II for 48 h. The cells were examined for the presentation of H-2Kb Ova peptide by FACS. One-way ANOVA with a Tukey post-hoc test was used for statistical analysis ($n = 3$/group, biological replicates). (E) The BMDCs were isolated from the syngeneic mouse and pulsed with ova peptide with or without macbecin II. TNF-α levels were examined in T cells co-cultured with BMDCs ($n = 3$/group, biological replicates). (F) E0771-Ova cells (500 cells/well) were seeded in a 96-well plate and treated with the indicated doses of macbecin II followed by co-culturing with T cells (E:T ratio 10:1) from (E) for 48 h. Dead cells were washed out with ice-cold PBS, and the cancer cells were fixed with methanol. and stained with crystal violet. The dye was then dissolved in 10% acetic acid, and readouts were obtained at 590 nm. Statistical significance between groups was determined using the one-way ANOVA with a Tukey post-hoc test ($n = 3$/group, biological replicates). Data are presented as mean $+/-$ SEM.

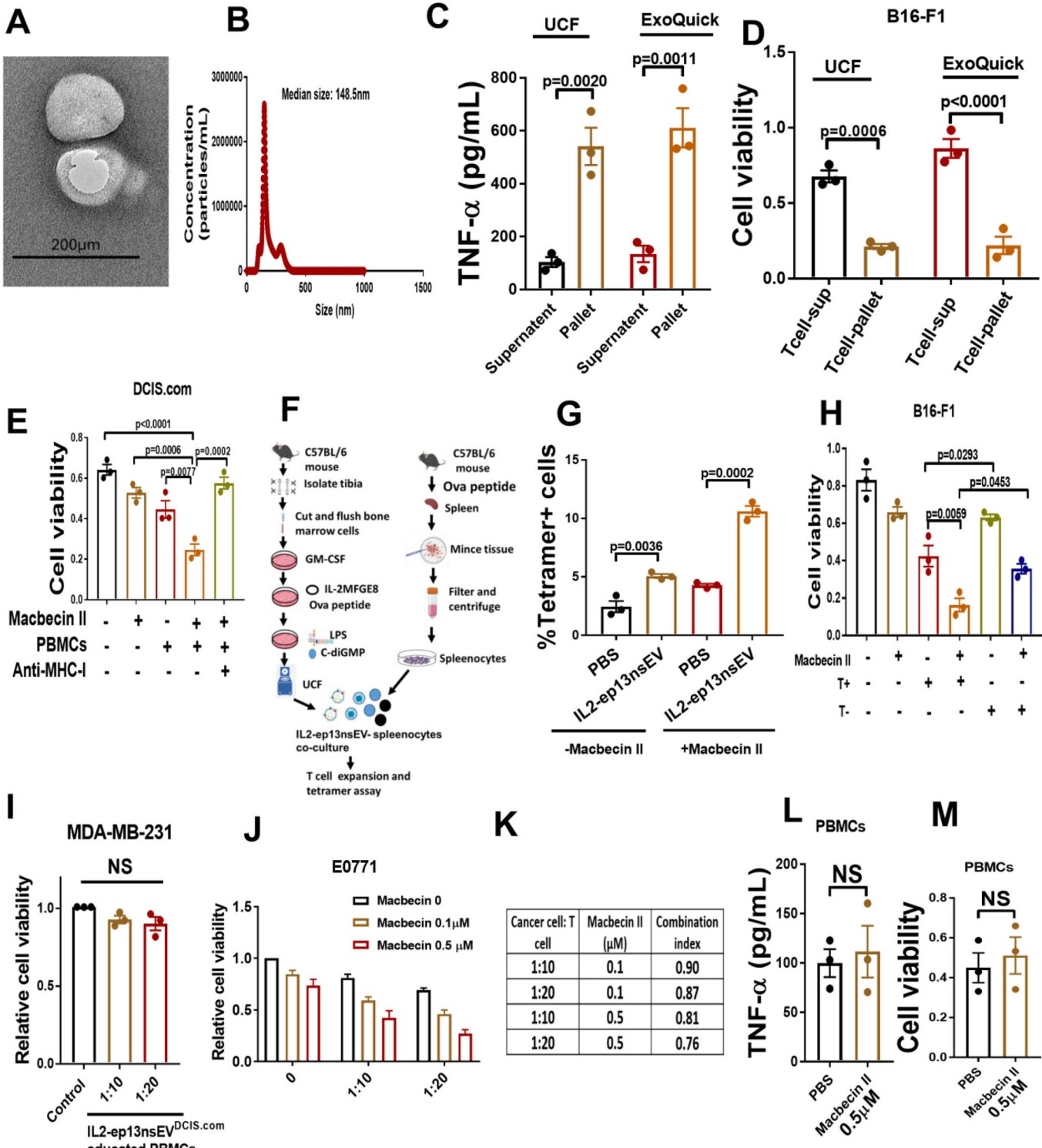

◄ **Figure EV3. Macbecin II potentiates anticancer efficacy of IL2-ep13nsEV in vitro.**

(A) Extracellular vesicles were isolated from dendritic cells that were pulsed with DCIS.com lysate. They were examined by electron microscopy. Representative image is shown. (B) The size distribution of p13nsEV was examined by Nanoparticle tracking analysis (NTA). The median size of p13nsEV was 148.5 nm. (C) The exosomes were isolated from the BMDCs pulsed with B16-F1 lysate by ultracentrifugation (UCF) or with the use of ExoQuick kit. The pallet or the supernatant fraction isolated from the UCF was then incubated with the T cells isolated from the spleen. Similarly, exosome fraction or exosome-depleted fraction from ExoQuick preparation was incubated with the T cells. The T cells were then co-cultured with the B16-F1 cells and TNF-α was examined ($n = 3$/group, biological replicates). The result was analyzed by the unpaired two-tailed Student's t-test ($n = 3$/group, biological replicates). (D) The T cells treated with the supernatant and pallet fraction were co-cultured with the B16-F1 cells (E:T ratio 15:1) for 48 h and relative cell viability was examined. After the incubation, the cells were washed with ice-cold PBS to remove dead cells. The live cells were fixed with methanol at room temperature for 15 min and then stained with crystal violet. The dye was dissolved in 10% acetic acid, and readouts were obtained at 590 nm. The result was analyzed by the unpaired two-tailed Student's t-test ($n = 3$/group, biological replicates). (E) DCIS.com cells (500 cells/well, $n = 3$/group, biological replicates) were seeded in a 96-well plate and co-cultured in the presence of PBMCs, macbecin II (0.1 μM), or 40 μg/mL MHC-I blocking antibody for 48 h. The IL2-ep13nsEV were prepared from the dendritic cells that were pulsed with DCIS.com lysate and they were used to educate T cells in the PBMC. After the incubation, the cells were washed with ice-cold PBS to remove dead cells. The live cells were fixed with methanol at room temperature for 15 min and then stained with crystal violet. The dye was dissolved in 10% acetic acid, and readouts were obtained at 590 nm. Statistical significance was determined using the one-way ANOVA with a Tukey post-hoc test. (F) Schematic of the experiment. BMDC were treated with GM-CSF for 5 days to differentiate monocytes into dendritic cells (DCs). The DCs were treated with ova peptide, LPS (100 ng/ml), C-diGMP (200 μM), and macbecin II followed by IL2-ep13nsEV isolation. The T cells were isolated from the spleen of syngeneic mice that were subcutaneously injected with ova peptide (5 μg for 10 days), with or without macbecin II, and they were activated in the presence of IL2-ep13nsEV. (G) Activated and expanded spleenocytes from (E) were examined for antigen-specific priming by tetramer assay. Unpaired Student's t-test was used for data analysis ($n = 3$/group/ biological replicates). (H) B16-F1 cells were seeded in a 96-well plate. Bone marrow-derived dendritic cells (BMDCs) isolated from syngeneic mice were pulsed with B16-F1 lysate and treated with or without macbecin II. Subsequently, T cells isolated from the spleen of syngeneic mouse were co-cultured with the BMDCs for antigenic priming. T cells co-cultured with macbecin II-treated, lysate-pulsed BMDCs were designated as T+, while those co-cultured with macbecin II-untreated BMDCs were designated as T−. The primed T cells (E:T ratio 10:1) were then co-cultured with B16-F1 cells treated with or without macbecin II for 48 h. After incubation, dead cells and T cells were washed off with PBS, and the remaining cancer cells were fixed with methanol and stained with crystal violet. The dye was dissolved in 10% acetic acid, and absorbance was measured at 590 nm. Statistical significance between groups was determined using one-way ANOVA with Tukey's post-hoc test ($n = 3$/group, biological replicates). (I) MDA-MB-231 cells (1000 cells/well, $n = 3$/group, biological replicates) were cultured in the presence of in vitro educated PBMCs from Fig. 3A for 48 h. Post-incubation, cells were washed with ice-cold PBS to dislodge the dead cells and PBMCs. The live cells were fixed in methanol at room temperature for 15 min and stained with crystal violet. The dye was dissolved in 10% acetic acid and readouts were obtained at 590 nm. (J) E0771 cells (500 cells/well) were cultured in a 96-well plate and treated with (i) vehicle, (ii) macbecin II (0.1 μM and 0.5 μM), (iii) CD8 T cells isolated from the spleen of an E0771-bearing syngeneic mouse and activated with exosomes isolated from E0771 lysate-loaded BMDCs (50 μg/mL) for 5 days (E:T ratio 5:1 and 10:1) alone or in combination for 48 h ($n = 3$/group, biological replicates). After incubation, the cells were washed with ice-cold PBS to remove dead cells and T cells. The surviving cells were fixed in methanol and stained with crystal violet. The dye was dissolved in 10% acetic acid and readouts were obtained at 590 nm. (K) Combination index (CI) values were calculated by CompuSyn based on the inhibitory effect of macbecin II and T cell treatment. CI < 1 indicates synergy. (L) TNF-α expressions were measured for PBMCs that were treated with macbecin II. (M) Relative cell viability was assessed using the MTS assay in PBMCs that were treated with macbecin II. ($n = 3$/group, biological replicates). Data are represented as mean +/− SEM ($n = 5$/group, biological replicates).

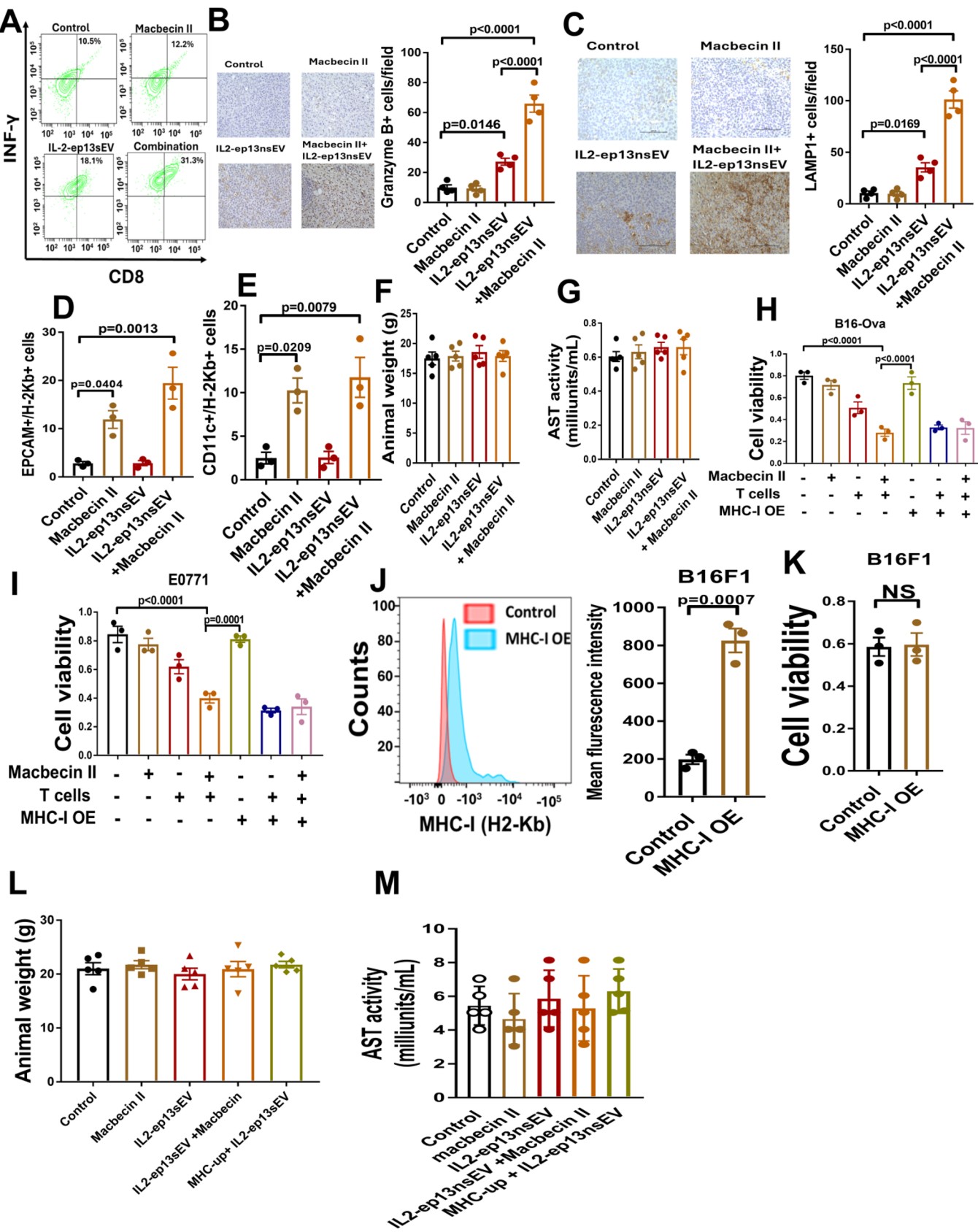

◀

**Figure EV4.  Macbecin II potentiates the anticancer efficacy of IL2-ep13nsEV in vivo.**

(**A**) Flow cytometry analysis for INF-γ-CD8 + T cells in the four groups. (**B**) Granzyme B expression was examined by immunohistochemistry (left) and quantified (right) in the four groups. The staining intensity was analyzed by the unpaired two-tailed Student's t-test ($n = 4$/group, biological replicates). (**C**) LAMP1 expression was examined through immunohistochemistry (left) and quantified (right) in the four groups. The staining intensity was measured and analyzed by the one-way ANOVA with a Tukey post-hoc test ($n = 4$/group, biological replicates). (**D, E**) MHC-I expression was examined in EPCAM+ tumor cells and CD11c+ dendritic cells in the dissociated tumor by FACS. Statistical analysis was performed using the one-way ANOVA with a Tukey post-hoc test ($n = 3$/group, biological replicates). (**F, G**) Animal weight (**F**) and AST activity (**G**) in the blood of mice were measured at the end-point. Data are represented as mean $+/-$ SEM ($n = 5$/group, biological replicates). (**H**) B16-Ova cells (500 cells/well, $n = 3$/group, biological replicates, were seeded in a 96-well plate and treated with a combination of macbecin II (0.1 μM) and OT-1 T cells (E:T ratio 5:1) with or without ectopic MHC-I expression for 48 h. The T cells were activated with IL2-ep13nsEV that were isolated from B16-Ova lysate-pulsed BMDCs (E:T ratio 5:1). After the incubation, cells were washed with ice-cold PBS to remove dead cells. The live cells were fixed with methanol at room temperature for 15 min and stained with crystal violet. The dye was dissolved in 10% acetic acid and measured at 590 nm. (**I**) E0771 cells (500 cells/well, $n = 3$/group, biological replicates) were seeded in a 96-well plate and treated with a combination of T cells (E:T ratio 5:1) and macbecin II (0.1 μM) with or without ectopic MHC-I expression for 48 h. The T cells were isolated from the spleen of E0771-bearing syngeneic mouse and they were activated with IL2-ep13nsEV that were isolated from E0771 lysate-pulsed BMDCs. After the incubation, cells were processed as described in (**H**). Statistical significance was determined by the one-way ANOVA with a Tukey post-hoc test. (**J**) MHC-I was ectopically expressed in B16-F1 cells, and the MHC-I expression was confirmed by FACS. The result was analyzed by the unpaired two-tailed Student's t-test ($n = 3$/group, biological replicates). (**K**) Cell viability was examined by MTS assay for B16-F1 with or without MHC-I over expression (OE) ($n = 3$/group, biological replicates). (**L, M**) Animal weight (**L**) and AST activity (**M**) were measured at the end-point ($n = 5$/group, biological replicates). Data are presented as mean $+/-$ SEM.

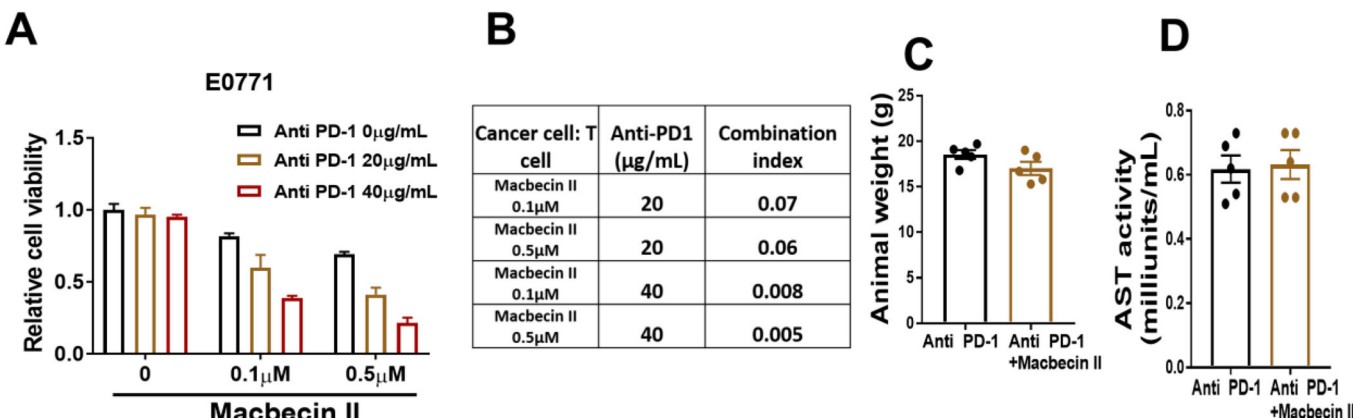

**Figure EV5. Macbecin II potentiates anticancer efficacy of anti-PD-1 immune checkpoint blockade in breast cancer.**

(**A**) E0771 cells (500 cells/well) were cultured in a 96-well plate in the presence of (i) vehicle, (ii) anti-PD-1 (20 μg and 40 μg), and (iii) Macbecin II (0.1 μM and 0.5 μM). CD8 T cells were isolated from the spleen of an E0771-bearing syngeneic mouse and activated with CD3/CD28 beads. The activated CD8 T cells were co-cultured with the E0771 cells ($n = 3$/group, biological replicates 5:1 E:T ratio). After 48 h, the cells were washed with ice-cold PBS to remove dead cells and T cells. The surviving cells were fixed in methanol and stained with crystal violet. The dye was dissolved in 10% acetic acid and readouts were obtained at 590 nm. (**B**) Combination index (CI) values were calculated by CompuSyn based on the inhibitory effect of anti-PD-1 and macbecin II treatment. CI < 1 indicates synergy. (**C, D**) Animal weight (**A**) and AST activity (**B**) were measured at the end-point ($n = 5$/group, biological replicates). Data are presented as mean $+/-$ SEM.

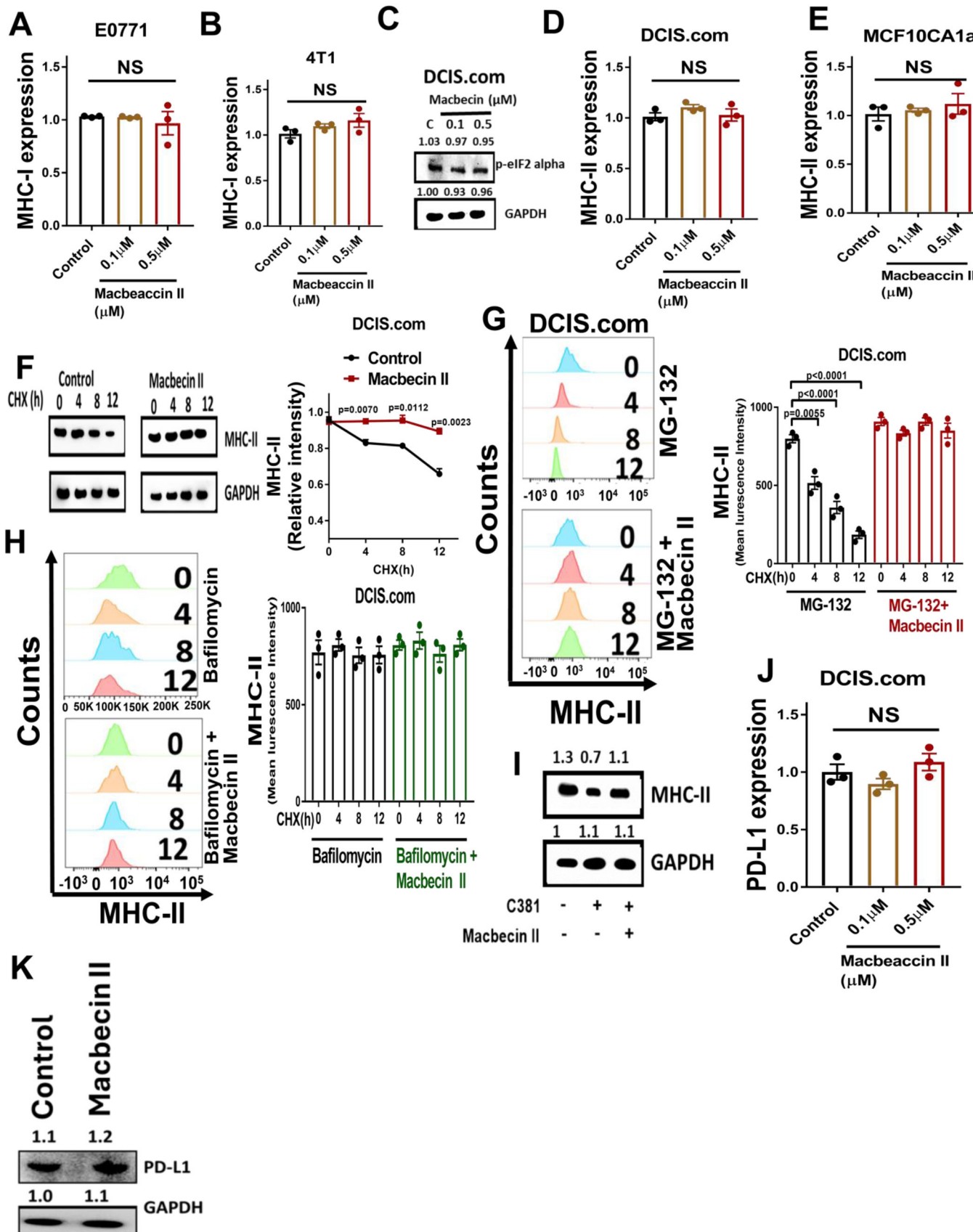

◀ **Figure EV6. Macbecin II upregulates MHC-I and II through inhibition of lysosomal degradation.**

(A, B) E0771 (A) and 4T1 (B) cells were seeded in a 24-well plate and treated with the indicated dose of macbecin II for 48 h. Total RNA was isolated, and cDNA synthesis was performed with the iScript cDNA Synthesis Kit as per the manufacturer's instructions followed by SYBR green-based Real Time PCR. GAPDH was used as an internal control. Statistical inference was determined by one-way ANOVA with a Tukey post-hoc test ($n = 3$/group, biological replicates). (C) DCIS.com cells were treated with the indicated dose of macbecin II for 48 h. Post-incubation, the total protein was isolated and p-eIF2 alpha expression was examined by western blot. Data are represented as mean $+/-$ SEM. (D, E) DCIS.com (D) and MCF10CA1a (E) cells were seeded in 24-well plates and treated with the indicated doses of macbecin II for 48 h. Total RNA was isolated, and cDNA synthesis was performed using the iScript™ cDNA Synthesis Kit according to the manufacturer's instructions, followed by SYBR Green-based real-time PCR. GAPDH was used as the internal control. Statistical analysis was performed using the one-way ANOVA with a Tukey post-hoc test ($n = 3$/group, biological replicates). (F) DCIS.com cells were treated with cycloheximide (50 μg/ml) and macbecin II (0.5 μM) for the indicated times. MHC-II expression was examined by western blot (left panel) which was quantified using ImageJ (right panel) ($n = 3$/group, biological replicates). A two-tailed unpaired Student's t-test was used for analysis. (G) DCIS.com cells were treated with vehicle, a combination of MG-132 (10 μM) and macbecin II (0.5 μM), or MG-132 alone in the presence of cycloheximide (50 μg/ml) at the indicated time. MHC-II expression was examined by FACS, and the data was analyzed by the one-way ANOVA with a Tukey post-hoc test ($n = 3$/group, biological replicates). (H) DCIS.com cells were treated with vehicle, a combination of Bafilomycin (100 nM) and macbecin II (0.5 μM), or Bafilomycin alone in the presence of cycloheximide (50 μg/ml) at the indicated time. MHC-II expression was then examined by FACS, and the data was analyzed by the one-way ANOVA with a Tukey post-hoc test ($n = 3$/group, biological replicates). (I) DCIS.com cells were seeded in a 24-well plate and cultured in the presence of cycloheximide (50 μg/ml). Cells were treated with the vehicle, a combination of C381 (30 μM) and macbecin II (0.5 μM), or C381 alone for 12 h and MHC-II expression was examined by western blot. (J) DCIS.com cells were seeded in a 24-well plate and treated with the indicated doses of macbecin II for 48 h. Total RNA was isolated, and cDNA synthesis was performed using the iScript™ cDNA Synthesis Kit according to the manufacturer's instructions, followed by SYBR Green-based Real-Time PCR. GAPDH was used as the internal control. Statistical analysis was performed using the one-way ANOVA with a Tukey post-hoc test ($n = 3$/group, biological replicates). (K) DCIS.com cells were treated with macbecin II (0.5 μM), and PD-L1 expression was examined by western blot. Data are presented as mean $+/-$ SEM.

