## [Peer Review File · EMBO Molecular Medicine]

MHC-I upregulation by macbecin II in the solid tumors potentiates the effect of active immunotherapy

Ravindra Despande, Kerui Wu, Shih-Ying Wu, Abhishek Tyagi, Eleanor Smith, Jee-Won Kim, and Kounosuke Watabe

Corresponding author(s): Kounosuke Watabe (kwatabe@wakehealth.edu)

Review Timeline:

Submission Date:	6th Aug 24
Editorial Decision:	3rd Sep 24
Revision Received:	19th Jan 25
Editorial Decision:	12th Feb 25
Revision Received:	19th Feb 25
Editorial Decision:	21st Feb 25
Revision Received:	21st Feb 25
Accepted:	28th Feb 25

Editor: Lise Roth

Transaction Report:

3rd Sep 2024

Dear Dr. Watabe,

Thank you for submitting your work to EMBO Molecular Medicine. We have now heard back from the referees who agreed to evaluate your manuscript. As you will see below, the reviewers raise substantial concerns on your work, which unfortunately preclude its publication in EMM in its current form.

The reviewers find that the question addressed by the study is of potential interest, however they remain unconvinced that some of the major conclusions are sufficiently supported by the data. They thus raise the following major issues:

- lack of evidence of an in vivo link between increased MHC class I expression and the efficacy of combined treatment
- lack of proper characterization of EV preparation

If you feel you can satisfactorily address these points and those listed by the referees, you may wish to submit a revised version of your manuscript. Regarding issue 1, referee #1 suggested transfecting the B16 line with H2b cDNA or using the B16-F1 clone which expresses high levels of MHC-I molecules. As these experiments might take time, we would be ready to extend the revision timeline to 6 months.

Please attach a covering letter giving details of the way in which you have handled each of the points raised by the referees. A revised manuscript will once again be subject to review, and we cannot guarantee at this stage that the eventual outcome will be favorable.

We are expecting your revised manuscript within six months, if you anticipate any delay, please contact us.

We require:

- 1) A .docx formatted version of the manuscript text (including legends for main figures, EV figures and tables). Please make sure that the changes are highlighted to be clearly visible.
- 2) Individual production quality figure files as .eps, .tif, .jpg (one file per figure). For guidance, download the 'Figure Guide PDF' (<https://www.embopress.org/page/journal/17574684/authorguide#figureformat>).
- 3) At EMBO Press we ask authors to provide source data for the main figures. Our source data coordinator will contact you to discuss which figure panels we would need source data for and will also provide you with helpful tips on how to upload and organize the files.
- 4) A .docx formatted letter INCLUDING the reviewers' reports and your detailed point-by-point responses to their comments. As part of the EMBO Press transparent editorial process, the point-by-point response is part of the Review Process File (RPF), which will be published alongside your paper.
- 5) A complete author checklist, which you can download from our author guidelines (<https://www.embopress.org/page/journal/17574684/authorguide#submissionofrevisions>). Please insert information in the checklist that is also reflected in the manuscript. The completed author checklist will also be part of the RPF.
- 6) All Materials and Methods need to be described in the main text using our 'Structured Methods' format, which is required for all research articles. According to this format, the Methods section includes a Reagents and Tools Table (listing key reagents, experimental models, software and relevant equipment and including their sources and relevant identifiers) followed by a Methods and Protocols section describing the methods using a step-by-step protocol format. The aim is to facilitate adoption of the methodologies across labs. More information on how to adhere to this format as well as a downloadable template (.docx) for the Reagents and Tools Table can be found in our author guidelines: <https://www.embopress.org/page/journal/17574684/authorguide#structuredmethods>
- 7) Please note that all corresponding authors are required to supply an ORCID ID for their name upon submission of a revised manuscript.
- 8) It is mandatory to include a 'Data Availability' section after the Materials and Methods. Before submitting your revision, primary

datasets produced in this study need to be deposited in an appropriate public database, and the accession numbers and database listed under 'Data Availability'. Please remember to provide a reviewer password if the datasets are not yet public (see <https://www.embopress.org/page/journal/17574684/authorguide#dataavailability>).

9) For data quantification: please specify the name of the statistical test used to generate error bars and P values, the number (n) of independent experiments (specify technical or biological replicates) underlying each data point and the test used to calculate p-values in each figure legend. The figure legends should contain a basic description of n, P and the test applied. Graphs must include a description of the bars and the error bars (s.d., s.e.m.). Please provide exact p values.

10) Our journal encourages inclusion of *data citations in the reference list* to directly cite datasets that were re-used and obtained from public databases. Data citations in the article text are distinct from normal bibliographical citations and should directly link to the database records from which the data can be accessed. In the main text, data citations are formatted as follows: "Data ref: Smith et al, 2001" or "Data ref: NCBI Sequence Read Archive PRJNA342805, 2017". In the Reference list, data citations must be labeled with "[DATASET]". A data reference must provide the database name, accession number/identifiers and a resolvable link to the landing page from which the data can be accessed at the end of the reference. Further instructions are available at .

11) We replaced Supplementary Information with Expanded View (EV) Figures and Tables that are collapsible/expandable online. A maximum of 5 EV Figures can be typeset. EV Figures should be cited as 'Figure EV1, Figure EV2" etc... in the text and their respective legends should be included in the main text after the legends of regular figures.

12) The paper explained: EMBO Molecular Medicine articles are accompanied by a summary of the articles to emphasize the major findings in the paper and their medical implications for the non-specialist reader. Please provide a draft summary of your article highlighting

13) Author contributions: CRedit has replaced the traditional author contributions section because it offers a systematic machine readable author contributions format that allows for more effective research assessment. Please remove the Authors Contributions from the manuscript and use the free text boxes beneath each contributing author's name in our system to add specific details on the author's contribution. More information is available in our guide to authors.

Please also suggest a visual abstract to illustrate your article as a PNG file 550 px wide x 300-600 px high. A cropped portion of this image will serve as thumbnail for the table of content on our webpage.

16) As part of the EMBO Publications transparent editorial process initiative (see our Editorial at <http://embomolmed.embopress.org/content/2/9/329>), EMBO Molecular Medicine will publish online a Review Process File (RPF) to accompany accepted manuscripts.

In the event of acceptance, this file will be published in conjunction with your paper and will include the anonymous referee

reports, your point-by-point response and all pertinent correspondence relating to the manuscript. Let us know whether you agree with the publication of the RPF and as here, if you want to remove or not any figures from it prior to publication. Please note that the Authors checklist will be published at the end of the RPF.

EMBO Molecular Medicine has a "scooping protection" policy, whereby similar findings that are published by others during review or revision are not a criterion for rejection. Should you decide to submit a revised version, I do ask that you get in touch after six months if you have not completed it, to update us on the status.

I look forward to receiving your revised manuscript.

Yours sincerely,

Lise Roth

***** Reviewer's comments *****

Referee #1 (Comments on Novelty/Model System for Author):

- The authors attempt to demonstrate the value of increasing MHC class I molecules, but do not analyse the anti-tumour CD8+T lymphocyte response in the various experiments.
- The in vivo models presented do not demonstrate that the efficacy of Macbecin II in combination with a vaccine or anti-PD-1 drug is dependent on an increase in MHC class I molecules.

Referee #1 (Remarks for Author):

The work on the activity of Macbecin II to increase MHC I expression as well as the mechanism is well presented. However, the relationship between the enhanced MHC class I expression on the efficacy of the combination of Macbecin II with cancer vaccine or anti-PD-1 is not demonstrated. As reported in the discussion Macbecin may have pleiotropic activities , although often at higher doses

Several complementary experiments are needed to validate the conclusions

- Since the mechanism of induction of MHC class I molecules is via the lysosomal pathway, does Macbecin II also increase MHC class II molecules?
- The authors need to use an anti-MHC-class I antibody in vitro to show that the increased activation of CD8+T cells is not a non-specific phenomenon (via IL-2 from the vaccine?).
- When the authors sensitize CD8+T cells in vitro with the vaccine, they do not show specific CD8+T cell induction. It is important to show that there is a specific priming of CD8+T cells in the different models to support the role of MHC-I molecule
- How is the in vitro T cell education performed? A priming period of 7 days is indicated. Usually several rounds of in vitro stimulation from naive CD8+T cells, are required to induce a specific CD8+T cell response.
- To show that the efficacy of the combination of Macbecin II with vaccine or anti-PD-1 is related to the increase of the MHC class I molecule and not to other activities of Macbecin II, can't the authors transfect a tumor line with this molecule (MHC class I) and show the loss of efficacy of the combination based on Macbecin II ?
- A second tumor model in in vivo experiments would increase the impact of this work.

Referee #2 (Comments on Novelty/Model System for Author):

Please see below my suggestions to replace melanoma model with breast cancer model, and to involve a transgenic breast cancer model as well.

Referee #2 (Remarks for Author):

In this manuscript, Deshpande RP et al screened a natural compound library to identify molecules that may improve the therapeutic efficiency in breast cancer. They show that macbecin II stimulates MHC-I expression, and it synergizes with dendritic cell (DC)-derived engineered extracellular vesicles (EV) as tumor vaccine and with anti-PD-1 immunotherapy. Of note, the authors had previously developed and described this IL2-ep13nsEV system and proved its effect in breast cancer. Deshpande RP et al also provide evidence that macbecin II acts via rescuing MHC-I from lysosomal degradation. The study uses molecular biological methods, in vitro cell culturing and major results are confirmed in in vivo experiments. The manuscript is generally well written, logical and easy to follow, and the application of macbecin II with tumor vaccines, such as specific EVs may open a novel possibility for further clinical trials. However, I have some concerns with the study.

Major concerns:

1. EVs were isolated with serial (ultra)centrifugation. How did the authors prove that this prepare lacked contaminating proteins that do not belong to EVs? They should provide data when using EV-depleted conditioned medium, and/or when EVs are isolated by an alternative method.
2. The characterization of the EV prepare according to MISEV2023 guidelines (<https://isevjournals.onlinelibrary.wiley.com/doi/full/10.1002/jev2.12416>) would be required. Could the authors show electron microscopic images, characterize the purity of the EV prepares, show a distribution of the EV size etc?
3. Although the manuscript focuses on breast cancer, the authors use B16-OVA mouse melanoma cells to examine the potential of macbecin II in eliciting an antigen-specific CD8 T cell response. This is surprising for the reviewer. Since tumor types may show a large difference in their genetic mutational spectrum, their epigenetic changes and phenotypic behaviour, the authors should carry out these experiments in a breast cancer model.
4. The authors use an implantation tumor model in these studies. However, this model does not represent the situation when tumor cells are endogeneously generated in the body. Thus, the authors should prove at least one of their key findings (e.g. the synergistic role of mecbacin II with EV-based tumor vaccine) in a transgenic breast cancer mouse model.

Minor comments:

5. „macbecin II synergistically reduced cancer cell death.....(Figure 3D,E)" -I guess it enhanced cell death or reduced cell viability as shown in Fig 3D.
6. „protein degradation was not affected by MG-132 (Figure 5D), while it was suppressed by bafilomycin (Figure 5E)" -Figure 5D and 5E show the opposite, there is discrepancy between the text and the figures.
7. Fig 4E,J,O,P: tissue images are very small, in the present form they do not provide any information. Could you please show larger images with a higher magnification?
8. Figure 4J,P: Could you please give a description of the H Score in the materials section? Could you please provide more details how tissues sections were evaluated? These should not be trivial for all the readers.
9. Figures show SEM. How many times were the experiments repeated, with how many biological parallels each time? Please include these parameters into all figure legends.
10. Do the authors have any hypothesis how macbecin II may modify lysosomal degradation? Although I understand that identifying this mechanism may be beyond the scope of the manuscript, however, it should be at least discussed in more details.

Referee #3 (Comments on Novelty/Model System for Author):

I believe this is an interesting study. They use adequate in vitro and mouse models to test their hypothesis. I am just missing a few controls to confirm the conclusions.

Referee #3 (Remarks for Author):

In this manuscript, Deshpande et al propose to use a low dose of macbecin II to improve immunotherapy in a model a breast cancer. They describe that macbecin II induces MHC-I upregulation through lysosomal degradation, which results in increased T cell activation and cytotoxic functions. Macbecin-induced MHC-I upregulation seem to work on both cancer cells and immune cells. This is an interesting study which proposes a new strategy to combine treatments to improve the response of breast cancer patients. The paper is clearly written and experiments are well performed and well controlled for the most parts.

Please find below some comments that could improve the manuscript:

- In Figure 2c, were naïve OTI cells used? I don't really understand the experimental set-up from the legend, text or methods. Naïve OTI cells usually need a few days (at least 3, usually 5 days) to become effector and produce large amounts of IFN γ . Do they also produce other cytokines such as IL2 and TNF α , given that polyfunctional T cells are seen are functionally superior?
- In addition, it is not clear whether OTI cells are more activated, expand more or are just better at producing cytokines. To directly test T cell activation, it would be important to look at early time points for CD69 up-regulation and proliferation.
- Finally, in Figure 2c, to formally show that macbecin II treatment of DCs leads to better T cell activation/effector function, I am missing the control of DC-OVA not treated with macbecin II.
- Figure 2D suggests that macbecin II leads to better OTI effector function. But for this to be accurate, the same number of OTI

cells should be added to B16-OVA cells. Is it the case? Overall, I would like a little more information, such as my question above, and the time for which OTI have been cultured with DCs before being co-cultured with B16-OVA.

- In Figure 2D, is the loss of B16-OVA viability due to IFN γ production or cytotoxic functions? Do the authors observe the same loss of viability if they block IFN γ and/or MHC-I?
- In Figure 3B, I am also missing the control showing that macbecin II-treated tumour cells have a superior capacity to induce IFN γ production by T cells compared to non-treated cells, and that this is dependent on antigen.
- In Figure 3C, has macbecin II any direct effect on the PBMC themselves (viability, activation)?
- In Figure 4H-I, are the cells restimulated *ex vivo*? It is often difficult to detect IFN γ -expressing T cells right out of the tumour, as T cells will secrete the IFN γ they produce extremely quickly if they see their cognate antigen. It would be useful to have more information on this, and a representative flow cytometry example. What about cytotoxic function (perforin/granzyme, LAMP-1 expression), is it also increased?
- In Figure 4J, which cells overexpress MHC-I? The authors described in previous figures that macbecin II induces MHC-I upregulation in both tumours and DCs, but it is unclear whether this is also the case *in vivo*, and whether up-regulating MHC-I on both cell types is required.
- In addition, in Figure 4J, I was surprised that EVs + macbecin II treatment exhibit increased MHC-I expression compared to macbecin II treatment only. Do the authors have an explanation for this?
- In Figure 5, does macbecin II also increase/stabilise MHC-II expression and immunoregulatory receptors (PD-1, etc...)? I wonder whether there is a widespread effect of macbecin II, especially given its mode of action, which could also contribute to its effect *in vivo*.

We thank reviewers for providing critical suggestions. We have addressed all critiques/questions point by point as summarized below and incorporated suggestions in the revised text, which we believe significantly improved the quality of the manuscript. The changes are underlined in the revised main manuscript file. Note that as per the editor's suggestion and in compliance with the journal format, the supplemental figures 1-5 are provided as Expanded View (EV) Figure 1-5 while the supplemental figure 6 is provided as Appendix Figure S1.

Referee #1:

General comment: The authors attempt to demonstrate the value of increasing MHC class I molecules, but do not analyze the anti-tumour CD8+T lymphocyte response in the various experiments.

Response: We thank reviewer for the concern. We have previously examined the CD8 T lymphocytes in response to IL2-ep13nsEV and macbecin II which promotes the MHC-I expression in breast cancer and found that IL2-ep13nsEV treatment enhanced primed CD4 and CD8 lymphocyte infiltration and activity (**please see the original Figure 4, F-I**) and also MHC-I expression (**please see Referee #1, Response #5 and the revised Figure 4, S, Referee #3, Response #9, revised Expanded View (EV) Figure 4, D**). To further examine the role of MHC-I in promoting the T cell infiltration, we performed an additional experiment in a melanoma model and found that macbecin II treatment promoted the CD4 and CD8 T cell infiltration as well as their activities. In addition, we found that the MHC-I upregulation phenocopied the effect of macbecin II in promoting the immune cell infiltration (**please see Referee #1, Response #5 and the revised Figure 4, O-R below**).

General comment: The in vivo models presented do not demonstrate that the efficacy of Macbecin II in combination with a vaccine or anti-PD-1 drug is dependent on an increase in MHC class I molecules.

Response: Macbecin II treatment was found to enhance the infiltration of CD4 and CD8 lymphocytes and upregulate MHC-I in the tumor when combined with anti PD-1 immune checkpoint blockade in breast cancer (**please see the original figure 5, E, F**). To delineate the significance of MHC-I, we overexpressed MHC-I in B16-ova and E0771 cells and treated the cancer cells with IL2-ep13nsEV-educated T cells and macbecin II. We found that the MHC-I overexpression phenocopied the effect of macbecin II in combination with IL2-ep13nsEV-educated T cells. (**please see Referee #1, Response #5 and the revised EV 4, H, I**).

Comment #1: Since the mechanism of induction of MHC class I molecules is via the lysosomal pathway, does Macbecin II also increase MHC class II molecules?

Response: We thank the reviewer for raising this concern. To address the question, we examined the cell surface expression of MHC II and found that, similar to MHC-I, macbecin II also upregulated the MHC-II expression. These results are incorporated in the **revised Expanded View (EV) Figure 1 (Figure EV1, C, D)**.

Figure EV1: Macbecin II stimulates MHC-I and II expression.

C: DCIS.com cells were treated with macbecin II at the indicated doses for 48 hours. The cells were trypsinized and stained with anti-MHC-II primary followed by secondary antibodies. Readouts were obtained using a BD FACS Canto. Mean fluorescence intensity was calculated and analyzed using the one-way ANOVA with a Tukey post-hoc test ($n = 3/\text{group}$, biological replicates). **D:** MCF10CA1a cells were treated with macbecin II at the indicated doses for 48 hours. The cells were processed as described in (C), and the results were analyzed using the one-way ANOVA with a Tukey post-hoc test ($n = 3/\text{group}$, biological replicates).

Accordingly, we have added the following description in the result section of the revised text: Similar to MHC-I, macbecin II treatment was found to upregulate MHC-II expression on the surface of DCIS.com (Figure EV1,C) and MCF10CA1a (Figure EV1,D) cells.

To examine the role of lysosomal degradation in mediating the MHC-II expression, we performed additional experiment. We treated the DCIS.com cells with proteosomal inhibitor MG-132 or lysosomal inhibitor bafilomycin in the presence of macbecin II and then examined the expression of cell surface MHC-II. We found that macbecin II upregulated MHC-II expression through the inhibition of lysosomal degradation. The results are presented below and incorporated in the revised Appendix Figure S1, G, H.

Appendix Figure S1: Macbecin II upregulates MHC-I and II through inhibition of lysosomal degradation.

G: DCIS.com cells were treated with vehicle, a combination of MG-132 (10 μM) and macbecin II (0.5 μM), or MG-132 alone in the presence of cycloheximide (50 $\mu\text{g}/\text{ml}$) at the indicated time. MHC-II expression was examined by FACS, and the data was analyzed by the one-way ANOVA with a Tukey post-hoc test ($n = 3/\text{group}$, biological replicates). **H:** DCIS.com cells were treated with vehicle, a combination of Bafilomycin (100 nM) and macbecin II (0.5 μM), or Bafilomycin alone in the presence of cycloheximide (50 $\mu\text{g}/\text{ml}$) at the indicated time. MHC-II expression was then examined by FACS, and the data was analyzed by the one-way ANOVA with a Tukey post-hoc test ($n = 3/\text{group}$, biological replicates).

We also examined the MHC-II expression on treatment with compound C381 which promotes the lysosomal acidification and found that, similar to MHC-I, the degradation of MHC-II was rescued by the macbecin II treatment. These results are incorporated in the **revised Appendix Figure S1, I**.

Appendix Figure S1: Macbecin II upregulates MHC-I and II through inhibition of lysosomal degradation.

I: DCIS.com cells were seeded in a 24-well plate and cultured in the presence of cycloheximide (50µg/ml). Cells were treated with the vehicle, a combination of C381 (30µM) and macbecin II (0.5µM), or C381 alone for 12h and MHC-II expression was examined by western blot.

Accordingly, we have added the following description in the result section of the revised text: Similar to MHC-I, we found that inhibiting proteasomal degradation did not affect the stability of MHC-II (Appendix Figure S1,G), while the lysosomal degradation inhibitor bafilomycin stabilized MHC-II expression (Appendix Figure S1, H). Additionally, the total expression of MHC-II protein was rescued when treated with C381 and macbecin II (Appendix Figure S1, I).

Comment #2: The authors need to use an anti-MHC-class I antibody *in vitro* to show that the increased activation of CD8+T cells is not a non-specific phenomenon (via IL-2 from the vaccine?).

Response: We previously showed that blockade of CD8 rescued the cell death when the cancer cells were co-cultured in the presence of PBMCs and treated with macbecin II (original Figure 3C). To confirm the role of MHC-I, we have performed an additional experiment and blocked the MHC-I by anti-MHC-I antibody as suggested by the reviewer. We found that blocking MHC-I rescued the cell death of DCIS.com cells that were co-cultured with PBMCs in the presence of macbecin II. The PBMCs used in the assay were educated with the IL2-ep13nsEV. We have added this results in the **revised Figure EV3,E** as shown below:

Figure EV3: Macbecin II potentiates anticancer efficacy of IL2-ep13nsEV *in-vitro*.

E: DCIS.com cells (500 cells/well, n = 3/group, biological replicates) were seeded in a 96-well plate and co-cultured in the presence of PBMCs, macbecin II (0.1µM), or 40 µg/mL MHC-I blocking antibody for 48 hours. The IL2-ep13nsEV were prepared from the dendritic cells that were pulsed with DCIS.com lysate and they were used to educate T cells in the PBMC. After the incubation, the cells were washed with ice-cold PBS to remove dead cells. The live cells were fixed with methanol at room temperature for 15 minutes and then stained with crystal violet. The dye was dissolved in 10% acetic acid, and readouts were obtained at 590 nm. Statistical significance was determined using the one-way ANOVA with a Tukey post-hoc test.

Accordingly, we have added the following description in the results section: Similar to blocking the CD8 T cells, inhibiting the MHC-II rescued the cell death of DCIS.com cells when they were co-cultured with IL2-ep13nsEV-educated PBMCs in the presence of macbecin II (Figure EV3.E).

Comment #3. When the authors sensitize CD8+T cells in vitro with the vaccine, they do not show specific CD8+T cell induction. It is important to show that there is a specific priming of CD8+T cells in the different models to support the role of MHC-I molecule

Response: To confirm that the specific priming of CD8 cells was dependent on MHC-I expression in a different model, we now performed an additional experiment. We isolated the splenocytes from the mouse and treated them with the IL2-ep13nsEV that were isolated from the ova-pulsed BMDCs. Then, the ova specific CD8 T cells in the splenocytes were examined by the ova tetramer assay. We found that the splenocytes treated with the ova specific IL2-ep13nsEV had more specific CD8 T cells when treated with macbecin II which upregulates MHC-I expression. These new results are incorporated in the **revised Figure EV3, F,G.**

Figure EV3: Macbecin II potentiates anticancer efficacy of IL2-ep13nsEV *in-vitro*.

F: Schematic of the experiment. BMDC were treated with GM-CSF for 5 days to differentiate monocytes into dendritic cells (DCs). The DCs were treated with ova peptide, LPS (100 ng/ml), C-diGMP (200 μ M), and macbecin II followed by IL2-ep13nsEV isolation. The T cells were isolated from the spleen of syngeneic mice that were subcutaneously injected with ova peptide (5 μ g for 10 days), with or without macbecin II, and they were activated in the presence of IL2-ep13nsEV. **G:** Activated and expanded splenocytes from **(F)** were examined for antigen specific priming by tetramer assay. Unpaired Student's t-test was used for data analysis (n = 3/group/ biological replicates).

Accordingly, we have added the following description in the text: To confirm the specific activation of T cells by the vaccine, we first pulsed BMDCs that were isolated from C57Bl/6 mice with OVA peptide, with or without macbecin II. Next, IL2-ep13sEV were prepared from BMDCs by ultracentrifugation. Subsequently, splenocytes were harvested from syngeneic mice that were inoculated with OVA peptide and treated with or without macbecin II. These splenocytes were then co-cultured with IL2-ep13sEV. Finally, CD8 T cells from the co-culture were examined for tetramer-positive cells (Figure EV3, F, G).

Comment #4: How is the in vitro T cell education performed? A priming period of 7 days is

indicated. Usually several rounds of in vitro stimulation from naive CD8+T cells, are required to induce a specific CD8+T cell response.

Response: We apologize for not providing the detailed information. We performed the activation of T cells with CD3/CD28 beads two times a week in the presence of IL2. In total, the T cells were activated and expanded for 2 weeks. They were then used in the subsequent experiments. We have now incorporated this description into the methods section.

Comment #5: To show that the efficacy of the combination of Macbecin II with vaccine or anti-PD-1 is related to the increase of the MHC class I molecule and not to other activities of Macbecin II, can't the authors transfect a tumor line with this molecule (MHC class I) and show the loss of efficacy of the combination based on Macbecin II ?

Response: We thank the reviewer for raising this concern. To show that the efficacy of the combination treatment is related to the increase of MHC-I expression, we ectopically expressed MHC-I in B16-F1 melanoma cells and then implanted the B16-F1 cells with or without MHC-I into syngeneic mice. The IL2-ep13nsEV was isolated from BMDC that were pulsed with B16-F1 lysate. The animals were then treated with IL2-ep13nsEV, macbecin II or combination of both. We found that the ectopic expression of MHC-I phenocopied the effect of macbecin II with the vaccine in reducing tumor growth, mitigating lung metastasis and promoting the immune cell infiltration. These results are added in the **revised Figure 4, J-T, Figure EV4, J-M**

Figure 4: Macbecin II potentiates the anticancer efficacy of IL2-ep13nsEV *in vivo*.

J-K: Schematic of the experiment. B16-F1 cells (2×10^5 cells) with or without ectopic MHC-I expression were implanted in syngeneic mice by subcutaneous injection. Animals were treated with the IL2-ep13nsEV alone or in combination with macbecin II at the indicated time and dose. **L:** Representative tumor images were taken at the endpoint (left). Tumor growth was monitored, and the result was analyzed by the two-way ANOVA ($n=10$ /group, biological replicates) test. Scale bar, 1cm. **M:** Tumor weight was measured at the endpoint and the result was analyzed by the one-way ANOVA with a Tukey post-hoc test ($n=10$ /group, biological replicates). **N:** Representative HE images showing lung micro-metastases from each group (left). Lung metastases were measured (right) and the result was analyzed by the one-way ANOVA ($n=5$ /group, biological replicates) test. Scale bar, $100\mu\text{m}$. **O-P:** CD8 (O) and CD4 (P) TILs among the CD3+ cells were measured in the dissociated tumor by FACS, and the result was analyzed by the one-way ANOVA with a Tukey post-hoc test ($n=4$ /group, biological replicates). **Q-R:** IFN- γ positive CD4 (Q) and CD8 (R) TILs were measured in the dissociated tumor by FACS, and the result was analyzed by the one-way ANOVA with a Tukey post-hoc test ($n=4$ /group, biological replicates). **S-T:** MHC-I expression was examined in tumor and DC cells in dissociated tumor by FACS and the result was analyzed by the one-way ANOVA with a Tukey post-hoc test ($n=3$ /group, biological replicates). Data are presented as mean \pm SEM (* $p < 0.05$, ** $p < 0.01$, **** $p < 0.0001$).

Figure EV4: Macbecin II potentiates anticancer efficacy of IL2-ep13nsEV *in-vivo*.

J: MHC-I was ectopically expressed in B16-F1 cells, and the MHC-I expression was confirmed by FACS ($n=3$ /group, biological replicates). **K:** Cell viability was examined by MTS assay for B16-F1 with or without MHC-I over expression (OE). Unpaired Student's T test was used to analyze the result ($n=3$ /group, biological replicates). **L,M:** Animal weight (L) and AST activity (M) were measured at the end-point, and the result was analyzed by the one-way ANOVA ($n=5$ /group, biological replicates). Data are presented as mean \pm SEM (* $p < 0.05$, ** $p < 0.01$, **** $p < 0.0001$).

In addition, we performed *in-vitro* experiment where we ectopically expressed mouse MHC-I in B16-ova and E0771 cell lines. We then isolated IL2-ep13nsEV from the BMDC that were pulsed with autologous tumor cell lysate, and they were incubated with the CD8 T cells that were isolated from spleen of syngeneic mouse. The cell viability was examined post co-culture. The results demonstrate that MHC-I overexpression phenocopied the effect of macbecin II resulting in loss of efficacy of combination treatment. These results are incorporated in the **revised Figure EV4, H,I**.

Figure EV4: Macbecin II potentiates anticancer efficacy of IL2-ep13nsEV *in-vivo*.

H: B16-Ova cells (500 cells/well, n = 3/group, biological replicates, were seeded in a 96-well plate and treated with a combination of macbecin II (0.1µM) and OT-1 T cells (E:T ratio 5:1) with or without ectopic MHC-I expression for 48h. The T cells were activated with IL2-ep13nsEV that were isolated from B16-Ova lysate-pulsed BMDCs (E:T ratio 5:1). After the incubation, cells were washed with ice-cold PBS to remove dead cells. The live cells were fixed with methanol at room temperature for 15 minutes and stained with crystal violet. The dye was dissolved in 10% acetic acid and measured at 590 nm. **I:** E0771 cells (500 cells/well, n = 3/group, biological replicates) were seeded in a 96-well plate and treated with a combination of T cells and macbecin II (0.1 µM) with or without ectopic MHC-I expression for 48h. The T cells were isolated from the spleen of E0771-bearing syngeneic mouse and they were activated with IL2-ep13nsEV that were isolated from E0771 lysate-pulsed BMDCs. After the incubation, cells were processed as described in (H). Statistical significance was determined by the one-way ANOVA with a Tukey post-hoc test.

Accordingly, we have added the following description in the text:

To further examine whether the enhanced efficacy of macbecin II in combination with IL-2-ep13nsEV is associated with increased MHC-I expression, we ectopically expressed mouse MHC-I in B16-Ova and E0771 cell lines. Subsequently, we isolated IL-2-ep13nsEV from BMDC pulsed with tumor cell lysate, and they were co-cultured with CD8 T cells that were isolated from the spleens of syngeneic mice. The T cells were then co-cultured with the cancer cells, and cell viability was assessed. Our results demonstrate that MHC-I overexpression reduced cell viability, with an effect comparable to macbecin II treatment (Figure EV4, H,I) when co-cultured in the presence of T cells. We further validated the efficacy of IL-2-ep13nsEV in sensitizing cell death in combination with MHC-I upregulation using an MHC-I-deficient (Peter *et al*, 2001; Seliger *et al*, 2001) B16-F1 melanoma model (Figure EV 4,J). The ectopic expression of MHC-I (Figure EV 4,J) did not affect the cell viability (Figure EV 4,K). IL-2-ep13nsEV^{B16F1} was generated from BMDCs that were pulsed with B16-F1 lysate (Figure 4J). B16-F1 cells, with or without MHC-I upregulation, were then implanted in syngeneic mice and they were treated with

macbecin II (2 mg/kg) and IL-2-ep13nsEV^{B16F1} (Figure 4K). Similar to the E0771 breast cancer model, we found that IL-2-ep13nsEV^{B16F1} alone and in combination with macbecin II significantly reduced tumor growth (Figure 4L), tumor weight (Figure 4M), and lung metastasis (Figure 4N). The combination of IL-2-ep13nsEV^{B16F1} and macbecin II enhanced the total intra-tumoral CD8+ (Figure 4O) and CD4+ (Figure 4P) populations, as well as the activated CD4+ (Figure 4Q) and CD8+ (Figure 4R) lymphocytes. The MHC-I upregulation phenocopied the effect of macbecin II treatment. Importantly, the treatment did not affect animal weight (Figure EV4, L) or serum AST activity (Figure EV4, M), indicating minimum toxic effects of the combination regimen. We also examined MHC-I protein expression on cancer cells and tumor-infiltrating dendritic cells. The combination of IL-2-ep13nsEV^{B16F1} and macbecin II significantly upregulated MHC-I on tumor (Figure 4S) and dendritic cells (Figure 4T). The MHC-I upregulation mimicked the effects of macbecin II in reducing tumor growth, mitigating lung metastasis, and enhancing CD4+/CD8+ lymphocyte infiltration.

Comment #6: A second tumor model in in vivo experiments would increase the impact of this work.

Response: We have now performed additional experiment using the B16F1 melanoma model and found that the combination of IL-2-ep13nsEV and macbecin II significantly reduced the tumor growth and promoted immune cell infiltration. The results are presented in #5 above (please see Fig 4J-T and Figure EV4 J-M).

Referee #2:

Major concerns:

Comment #1: EVs were isolated with serial (ultra)centrifugation. How did the authors prove that this prepriate lacked contaminating proteins that do not belong to EVs? They should provide data when using EV-depleted conditioned medium, and/or when EVs are isolated by an alternative method.

Response: We thank the reviewer for raising this concern. To address it, we first isolated BMDCs from syngeneic mice and pulsed them with B16-F1 lysate. Exosomes were then isolated by ultracentrifugation. Following this, T cells isolated from the spleen of syngeneic mouse were treated with the supernatant or the pellet fractions obtained from ultracentrifugation. We found that T cells significantly secreted TNF- α when treated with the pellet fraction, which contains exosomes, compared to the EV-depleted supernatant. Additionally, we performed a cell viability assay with T cells and found that exosome-primed T cells considerably reduced cell viability compared to supernatant-treated T cells.

Furthermore, we isolated exosomes using the ExoQuick kit and treated T cells with the exosome fraction or the exosome-depleted medium. Similar to the results obtained with ultracentrifugation, the exosome fraction isolated using the ExoQuick kit more effectively activated T cells and reduced cell viability after co-culture with cancer cells. These results have been included in the revised **Figure EV3, C and D**.

Figure EV3: Macbecin II potentiates anticancer efficacy of IL2-ep13nsEV in-vitro.

C: The exosomes were isolated from the BMDCs pulsed with B16-F1 lysate by ultracentrifugation (UCF) or with the use of ExoQuick kit. The pallet or the supernatant fraction isolated from the UCF was then incubated with the T cells isolated from the spleen. Similarly, exosome fraction or exosome-depleted fraction from ExoQuick preparation was incubated with the T cells. The T cells were then co-cultured with the B16-F1 cells and TNF- α was examined. (n=3/group, biological replicates). **D:** The T cells treated with the supernatant and pallet fraction were co-cultured with the B16-F1 cells (E:T ratio 15:1) for 48h and relative cell viability was examined. After the incubation, the cells were washed with ice-cold PBS to remove dead cells. The live cells were fixed with methanol at room temperature for 15 minutes and then stained with crystal violet. The dye was dissolved in 10% acetic acid, and readouts were obtained at 590 nm. The result was analyzed by the unpaired two-tailed Student's t-test (n=3/group, biological replicates). Data are represented as mean +/- SEM. (*p<0.05, **p<0.01, ****p<0.0001).

Comment #2: The characterization of the EV prepare according to MISEV2023 guidelines (<https://isevjournals.onlinelibrary.wiley.com/doi/full/10.1002/jev2.12416>) would be required. Could the authors show electron microscopic images, characterize the purity of the EV preparates, show a distribution of the EV size etc?

Response: We have confirmed the nanoparticle size and distribution by electron microscopy and nanoparticle tracking analysis. The results are incorporated in the **revised Figure EV3, A,B**

Figure EV3: Macbecin II potentiates anticancer efficacy of IL2-ep13nsEV *in-vitro*.

A: Extracellular vesicles were isolated from dendritic cells that were pulsed with DCIS.com lysate. They were examined by electron microscopy. Representative image is shown. **B:** The size distribution of p13nsEV was examined by Nanoparticle tracking analysis (NTA). The median size of p13nsEV was 148.5 nm.

Accordingly, the following description was added in the methods section:
 Analysis of extracellular vesicles by transmission electron microscopy:
 The extracellular vesicles were visualized using negative staining through electron microscopy. The vesicles were purified and placed onto discharged 200-mesh copper EM grids, followed by fixation with 2% paraformaldehyde (PFA). A 1% uranyl acetate solution was used to stain the extracellular vesicles on the EM grids, which were then imaged using an FEI Tecnai BioTwin Transmission Electron Microscope.

We have added the following description in the text:
 We confirmed the size and distribution of extracellular vesicles by nanoparticle tracking and electron microscopy (Figure EV3,A, B). Additionally, we showed that the exosome enriched fraction activated the T cells more significantly as compared with the exosome depleted supernatant (Figure EV3,C,D).

Comment #3: Although the manuscript focuses on breast cancer, the authors use B16-OVA mouse melanoma cells to examine the potential of macbecin II in eliciting an antigen-specific CD8 T cell response. This is surprising for the reviewer. Since tumor types may show a large difference in their genetic mutational spectrum, their epigenetic changes and phenotypic behaviour, the authors should carry out these experiments in a breast cancer model.

Response: We have repeated the experiment using the mouse breast cancer cell line E0771. Ova protein was ectopically expressed in E0771 cells, and the H2-Kb-Ova presentation was examined following treatment with macbecin II. To examine the antigen specific response, we first isolated the BMDCs from syngeneic mouse and pulsed the DCs with ova. These DCs were then used to educate T cells isolated from the spleen of C57 mouse. T cell activity was assessed by measuring TNF- α secretion in the supernatant. We found that TNF- α is significantly secreted from the BMDCs that were pulsed with ova and treated with macbecin II. The functional activity of the expanded T cells was further evaluated by co-culturing them with E0771 cells. The combination of T cells and macbecin II significantly reduced the viability of E0771-Ova cells. These results demonstrate that macbecin II enhances antigen-specific T cell responses. The results have been incorporated into the **revised Figure EV2, D–F**.

Figure EV2: Macbecin II promotes antigen presentation and enhances antigen-dependent cancer cell death.

D: E0771 cells transfected with the ova plasmid were treated with the indicated doses of macbecin II for 48 hours. The cells were examined for the presentation of H2-Kb ova peptide by FACS. One-way ANOVA with a Tukey post-hoc test was used for statistical analysis ($n = 3/\text{group}$, biological replicates). **E:** The BMDCs were isolated from the syngeneic mouse and pulsed with ova peptide with or without macbecin II. TNF- α levels were examined in T cells co-cultured with BMDCs ($n = 3/\text{group}$, biological replicates). **F:** E0771-Ova cells (500 cells/well) were seeded in a 96-well plate and treated with the indicated doses of macbecin II followed by co-culturing with T cells (E: T ratio 10:1) from (E) for 48 hours. Dead cells were washed out with ice cold PBS, and the cancer cells were fixed with methanol. and stained with crystal violet. The dye was then dissolved in 10% acetic acid, and readouts were obtained at 590 nm. Statistical significance between groups was determined using the one-way ANOVA with a Tukey post-hoc test ($n = 3/\text{group}$, biological replicates). Data are presented as mean \pm SEM (* $p < 0.05$, ** $p < 0.01$, **** $p < 0.0001$).

To further confirm the role of macbecin II in inducing antigen-dependent cell death, we first ectopically expressed ova in murine breast cancer cells (E0771) and then treated with macbecin II. We found that macbecin II induced the cell surface expression of H2-Kb-ova. (Figure EV 2,D). The BMDCs were isolated from spleen of syngeneic mouse and pulsed with ova peptide. The ova-pulsed BMDCs were then treated with or without macbecin II and used for co-culture with the T cells. TNF- α secretion was examined in the supernatant as a marker of T cell activation. We found that the ova-pulsed and macbecin II treated BMDCs significantly promoted TNF- α secretion from the T cells. (Figure EV 2,E). Additionally, we co-cultured these T cells with E0771-Ova cells in the presence of macbecin II, and found that they significantly reduced tumor cell viability (Figure EV 2,F).

Comment #4: The authors use an implantation tumor model in these studies. However, this model does not represent the situation when tumor cells are endogenously generated in the body. Thus, the authors should prove at least one of their key findings (e.g. the synergistic role of mecbacin II with EV-based tumor vaccine) in a transgenic breast cancer mouse model.

Response: We thank the reviewer for raising this concern. To address this question, we used B6.FVB-Tg(MMTV-PyVT)634Mul/LellJ mouse which endogenously forms the mammary tumors. The animals were treated with anti PD-1 immune checkpoint blockade alone or in combination with macbecin II. We found that the combination treatment significantly reduced tumor growth. These results are now incorporated in the **revised figure 5,G-I**.

Figure 5: Macbecin II potentiates anticancer efficacy of anti PD-1 immune checkpoint blockade in breast cancer.

G: Representative tumor images were taken at the endpoint (left). Tumor growth was monitored, and the result was analyzed by the two-way ANOVA test (n=5/group, biological replicates). Scale bar, 1cm. **H:** Average tumor weight was measured at the endpoint and the result was analyzed by the unpaired two-tailed Student's t-test (n=5/group, biological replicates). **I:** Body weight of mice was measured at the end-point and the result was analyzed by the unpaired two-tailed Student's t-test (n=5/group, biological replicates). Data are represented as mean +/- SEM. (n=5/group, biological replicates). (*p<0.05, **p<0.01, ****p<0.0001).

Accordingly, we have added the following description in the text:

To further examine the role of macbecin II in potentiating the effect of anti PD-1 immune checkpoint blockade in the endogenously formed tumors, we used genetically modified B6.FVB-Tg(MMTV-PyVT)634Mul/LelJ mouse which spontaneously forms the mammary tumors. The animals were treated with macbecin II alone or in combination with anti PD-1 immune checkpoint blockade when the tumors became palpable. We found that macbecin II potentiated the effect of anti PD-1 immune checkpoint blockade in reducing the tumor growth (Figure 5, G, H) without affecting the body weight (Figure 5,I).

Minor comments:

Comment #5: „macbecin II synergistically reduced cancer cell death.....(Figure 3D,E)" -I guess it enhanced cell death or reduced cell viability as shown in Fig 3D.

Response: We have corrected the sentence as: 'macbecin II synergistically reduced cancer cell viability..'

Comment #6: „protein degradation was not affected by MG-132 (Figure 5D), while it was suppressed by bafilomycin (Figure 5E)" -Figure 5D and 5E show the opposite, there is discrepancy between the text and the figures.

Response: We are sorry for this error. We have now corrected the statement in the text. The correct statement is: The protein degradation mediated by MG-132 treatment was rescued by the macbecin II (Figure 6D) treatment, while the lysosomal degradation inhibitor bafilomycin suppressed the MHC-I degradation (Figure 6E).

Comment #7: Fig 4E,J,O,P: tissue images are very small, in the present form they do not provide any information. Could you please show larger images with a higher magnification?

Response: We have provided the images with higher magnification for the original Figure 4 and 5.

Comment #8: Figure 4J,P: Could you please give a description of the H Score in the materials section? Could you please provide more details how tissues sections were evaluated? These should not be trivial for all the readers.

Response: We have now provided the details in the Methods section on how we calculated the H score in detail. The description included as follow:

IHC quantification: The stained cells were quantified to calculate the H score manually. Cells stained were categorized as: weak, moderate and strong. Then, H score was calculated with the formula $(1 \times \text{percentage of weak staining}) + (2 \times \text{percentage of moderate staining}) + (3 \times \text{percentage of strong staining})$. The observer was blinded for the group of samples used.

Comment #9: Figures show SEM. How many times were the experiments repeated, with how many biological parallels each time? Please include these parameters into all figure legends.

Response: The experiments were repeated 3 times (biological replicates). We have now included these details in the figure legends.

Comment #10: Do the authors have any hypothesis how macbecin II may modify lysosomal degradation? Although I understand that identifying this mechanism may be beyond the scope of the manuscript, however, it should be at least discussed in more details.

Response: Macbecin II may possibly modify the lysosomal degradation through the following mechanisms: (i) Inhibiting the translocation of MHC-I in lysosomes. (ii) Increasing the lysosomal acidification. (iii) Macbecin II may inhibit RAB7b and /or promote RAB11 to increase the membrane MHC-I localization. We have included this statement in the Discussion section.

Referee #3:

Comment #1: In Figure 2c, were naïve OTI cells used? I don't really understand the experimental set-up from the legend, text or methods. Naïve OTI cells usually need a few days (at least 3, usually 5 days) to become effector and produce large amounts of IFNg. Do they also produce other cytokines such as IL2 and TNFa, given that polyfunctional T cells are seen are functionally superior?

Response: We thank the reviewer for raising this concern. We have now updated the details in the figure legend. The OT-1 T cells were activated and expanded with CD3/28 beads for 5 days after the co-culture with the ova-pulsed DCs and then used for INF- γ staining. As suggested, we also examined the TNF- α in the conditioned medium after co-culture with the B16-Ova cells. We have incorporated the results in the **revised Figure EV2,B**.

Figure EV2: Macbecin II promotes antigen presentation and enhances antigen-dependent cancer cell death.
B: OT-1 cells were co-cultured with B16-Ova cells and the expression of TNF- α in the medium was measured by ELISA. Statistical significance was determined using one-way ANOVA with a Tukey post-hoc test (n = 3/group, biological replicates).

We have added the following description in the text:

Activation of OT-1 T cells was assessed by measuring IFN- γ expression with FACS (Figure 2C) and TNF- α levels (Figure EV2,B) by ELISA. Our results demonstrated significant activation of OT-1 T cells when they were co-cultured with ova-pulsed DCs.

Comment #2: In addition, it is not clear whether OTI cells are more activated, expand more or are just better at producing cytokines. To directly test T cell activation, it would be important to look at early time points for CD69 up-regulation and proliferation.

Response: We have previously activated and expanded the OT-1 T cells after co-culture with the ova-pulsed DCs for 5 days before using it for the co-culture. As suggested, we also examined the CD69 expression by FACS and found that it is increased after co-culture with the DCs. We have incorporated the results in the **revised Figure EV2, C**.

Figure EV2: Macbecin II promotes antigen presentation and enhances antigen-dependent cancer cell death.

C: OT-1 T cells were co-cultured in the presence of DCs that were pulsed with Ova (DC-ova) with or without macbecin II, and then examined for CD69 surface expression by FACS at after three days. Statistical significance was determined using the one-way ANOVA with a Tukey post-hoc test (n = 3/group, biological replicates).

Accordingly, the following description was added in the text:

Macbecin II was found to significantly enhance lymphocyte activation marker CD69 expression on OT-1 T cells at Day 3 (Figure EV2, C).

Comment #3: Finally, in Figure 2c, to formally show that macbecin II treatment of DCs leads to better T cell activation/effector function, I am missing the control of DC-OVA not treated with macbecin II.

Response: We have now included the control DC-ova not treated with macbecin to show that macbecin II treatment of DCs leads to better T cell activation. We have incorporated the results in the **revised Figure 2C**.

Figure 2: Macbecin II promotes antigen presentation and enhances antigen dependent cancer cell death.

C: The OT-1 T cells were co-cultured with DC in the presence of Macbecin II or PBS, and they were examined for IFN- γ expression by FACS as described in (B). Statistical significance was analyzed by the one-way ANOVA with a Tukey post-hoc test (n = 3/group, biological replicates).

Comment #4: Figure 2D suggests that macbecin II leads to better OTI effector function. But for this to be accurate, the same number of OTI cells should be added to B16-OVA cells. Is it the case? Overall, I would like a little more information, such as my question above, and the time for

which OT1 have been cultured with DCs before being co-cultured with B16-OVA.

Response: We apologize for not providing the detailed information. We have co-cultured B16-Ova cells with the same number of OT-1 T cells (3rd bar, original Figure 2D). We first co-cultured the OT-1 T cells with Ova-DCs for a week followed by activation and expansion with CD3/CD28 microbeads for 5 days. The activated OT-1 T cells were then used for the co-culture with the B16-ova cells.

Comment #5: In Figure 2D, is the loss of B16-OVA viability due to IFN γ production or cytotoxic functions? Do the authors observe the same loss of viability if they block IFN γ and/or MHC-I?

Response: We now performed rescue experiment by blocking the MHC-I expression and found that it rescued the reduced cell viability. These results are incorporated in the **revised Figure 2D**.

Figure 2: Macbecin II promotes antigen presentation and enhances antigen dependent cancer cell death. **D:** B16-Ova cells (500 cells/well) were seeded into a 96-well plate and treated with the macbecin II (0.5 μ M), OT-1 T cells (E:T ratio 10:1), and MHC-I blocking antibody (40 μ g/mL) for 48 hours. Dead cells and OT-1 T cells were washed off with PBS, and the remaining cancer cells were fixed with methanol and stained with crystal violet. The dye was then dissolved in 10% acetic acid, and the absorbance was measured at 590 nm. Statistical significance between groups was determined using the one-way ANOVA with a Tukey post-hoc test (n = 3/group, biological replicates). Data are presented as mean +/-SEM (*p < 0.05, **p < 0.01, ****p < 0.0001).

Accordingly, the following description was added in the text:

Furthermore, we evaluated the efficacy of activated OT-1 cells in mediating antigen-dependent killing of target cells by co-culturing OT-1 T cells with B16-Ova cells in the presence of macbecin II. Our results revealed that OT-1 T cells exerted a significant cytotoxic effect on B16-Ova cells, which was dependent on MHC-I expression (Figure 2D).

Comment #6: In Figure 3B, I am also missing the control showing that macbecin II-treated tumour cells have a superior capacity to induce IFN γ production by T cells compared to non-treated cells, and that this is dependent on antigen.

Response: We thank the reviewer for raising this concern. We have now repeated the experiment. First, we educated the T cells in the PBMCs with IL2-ep13nsEV. Next, the PBMCs were co-cultured with DCIS.com cells that were treated with or without macbecin II. The INF- γ + CD8 T cells were subsequently analyzed by FACS. These results are incorporated into the **revised Figure 3B**

Figure 3: Macbecin II potentiates anticancer efficacy of IL2-ep13nsEV tumor vaccine.
B: PBMCs from (A) were co-cultured with the DCIS.Com cells treated with or without macbecin II (0.5μM). Followed by this, IFN-γ+ CD8 T cell were examined by FACS. Statistical significance was analyzed by the one-way ANOVA with a Tukey post-hoc test (n = 3/group, biological replicates).

Comment #7: In Figure 3C, has macbecin II any direct effect on the PBMC themselves (viability, activation)?

Response: We have now examined the effect of macbecin II on T cell viability and activation and found that there was insignificant difference in cell viability and TNF-α activity between control and macbecin II treated T cells. These results are incorporated in the **revised Figure EV3, L,M.**

Figure EV3: Macbecin II potentiates anticancer efficacy of IL2-ep13nsEV *in vitro*.
L: TNF-α expressions were measured for PBMCs that were treated with macbecin II.
M: Relative cell viability was assessed using the MTS assay in PBMCs that were treated with macbecin II. Statistical analyses were performed using the unpaired Student's t-test (n = 3/group, biological replicates).

Therefore, we have added the following description in the text:
 Our results indicated that macbecin II did not affect the activity or viability of PBMCs (Figure EV3, L,M), suggesting that macbecin II primarily targets cancer cells to induce cell death through enhanced MHC-I expression.

Comment #8: In Figure 4H-I, are the cells restimulated ex vivo? It is often difficult to detect IFNγ-expressing T cells right out of the tumour, as T cells will secrete the IFNγ they produce extremely quickly if they see their cognate antigen. It would be useful to have more information on this, and a representative flow cytometry example. What about cytotoxic function (perforin/granzyme, LAMP-1 expression), is it also increased?

Response: We thank the reviewer for this question. We have now added the description in the

methods section as below:

To examine the tumor infiltrating lymphocytes, the tumor tissues were minced and incubated with the tissue dissociation solution (1mg/mL collagenase IV, RPMI 1650, 5% FBS, 200U/mL, DNase) at 37°C for 20 minutes. The cells were passed through a 70µm strainer, and RBC lysis was performed with RBC lysis buffer. The single cells were then incubated with 100ng/ml PMA and 1µg/ml ionomycin and 2.5 mg/ml protein transport inhibitor Brefeldin A at 37°C for 4 hours. The dead cells were then stained using the Zombie Aqua Fixable Viability Kit. The cells were then washed with PBS and analyzed or further fixed in 4% PFA in PBS for 15 min for intracellular cytokine staining. For intracellular cytokine staining, cells were treated with permeabilization buffer followed by staining with antibodies for 30 minutes. The cells were then examined by the BD Canto II Flow Cytometer, and the data were analyzed by FlowJo software.

Accordingly, we have incorporated the following result of FACS for CD8 T cells in the **revised Figure EV4, A**.

Figure EV4: Macbecin II potentiates the anticancer efficacy of IL2-ep13nsEV *in vivo*.
A: Flow cytometry analysis for INF-γ -CD8+ T cells in the four groups.

Furthermore, we examined the granzyme B and LAMP1+ cells in the four groups and found that the positively stained cells were significantly higher in the combination treatment group. The results are presented below and incorporated the **in revised figure EV4, B,C**.

Figure EV4: Macbecin II potentiates the anticancer efficacy of IL2-ep13nsEV *in vivo*.
B: Granzyme B expression was examined by immunohistochemistry (left) and quantified (right) in the four groups. The staining intensity was analyzed by the unpaired two-tailed Student's t-test (n=4/group, biological replicates). **C:** LAMP1 expression was examined through immunohistochemistry (left) and quantified (right) in the four groups. The staining intensity was measured and analyzed by the one-way ANOVA with a Tukey post-hoc test (n=4/group, biological replicates).

Accordingly, we have added the following description in the text:

The combination treatment significantly increased the number of cells that expressed Granzyme B (Figure EV4,B) and LAMP1 (Figure EV4,C) in animals treated with combination of macbecin II and IL2-ep13sEV.

Comment #9: In Figure 4J, which cells overexpress MHC-I? The authors described in previous figures that macbecin II induces MHCI upregulation in both tumours and DCs, but it is unclear whether this is also the case *in vivo*, and whether up-regulating MHC-I on both cell types is required.

Response: We thank the reviewer for this concern. We now performed an additional animal experiment as depicted in the original Figure 4A-B and examined the MHC-I expression among the tumor cells and the dendritic cells. We found that the expression of MHC-I was increased on both cell types. These results are incorporated in the **revised Figure EV4, D, E**.

Figure EV4: Macbecin II potentiates the anticancer efficacy of IL2-ep13nsEV *in vivo*. **D-E:** MHC-I expression was examined in EPCAM+ tumor cells and CD11c+ dendritic cells in the dissociated tumor by FACS. Statistical analysis was performed using the one-way ANOVA with a Tukey post-hoc test (n = 3/group, biological replicates).

Accordingly, the following description was added in the text:

We also examined MHC-I protein expression and found that the combination of IL-2-ep13nsEV and macbecin II significantly promoted MHC-I expression on both dendritic and cancer cells (Figure EV4, D,E).

We have demonstrated that macbecin II promotes an antigen-specific response (**Reviewer #1, Comment #3; revised Figure EV3, F, G**). Upregulating MHC-I expression on cancer cells was found to induce cell death when co-cultured with T cells (**Reviewer #1, Comment #5; revised Figure EV4, H, I; Reviewer #2, Comment #3; revised Figure EV2, D,F**). Similarly, we examined the impact of macbecin II on the antigen-presenting potential of dendritic cells (DCs). We found that macbecin II treatment significantly activated T cells, as indicated by higher TNF- α secretion (**Reviewer #2, Comment #3; revised Figure EV2, E. Reviewer #3, Comment #1; revised Figure EV2, B, and Reviewer #3, Comment #3; revised Figure 2C**).

To address the concern regarding whether MHC expression on dendritic cells or cancer cells is required for a better response, we conducted an additional experiment. We isolated bone marrow-derived dendritic cells (BMDCs) from syngeneic mice. The BMDCs were pulsed with B16-F1 cell lysate in the presence or absence of macbecin II. BMDCs treated with or without macbecin II and lysate were co-cultured with T cells isolated from the spleen of syngeneic mouse. T cells co-cultured with macbecin II-treated, lysate-pulsed BMDCs were designated as

T+, while those co-cultured with macbecin II-untreated BMDCs were designated as T-. Subsequently, B16-F1 cells were treated with or without macbecin II to upregulate MHC-I expression, followed by co-culture with the T cells. We found that MHC-I expression on both cancer cells and dendritic cells was essential to exert the cytotoxic effect. This result is incorporated into **revised Figure EV3,H**.

Figure EV3: Macbecin II potentiates anticancer efficacy of IL2-ep13nsEV *in-vitro*.
H: B16-F1 cells were seeded in a 96-well plate. Bone marrow-derived dendritic cells (BMDCs) isolated from syngeneic mice were pulsed with B16-F1 lysate and treated with or without macbecin II. Subsequently, T cells isolated from the spleen of syngeneic mouse were co-cultured with the BMDCs for antigenic priming. T cells co-cultured with macbecin II-treated, lysate-pulsed BMDCs were designated as T+, while those co-cultured with macbecin II-untreated BMDCs were designated as T-. The primed T cells (E:T ratio 10:1) were then co-cultured with B16-F1 cells treated with or without macbecin II for 48 hours. After incubation, dead cells and T cells were washed off with PBS, and the remaining cancer cells were fixed with methanol and stained with crystal violet. The dye was dissolved in 10% acetic acid, and absorbance was measured at 590 nm. Statistical significance between groups was determined using one-way ANOVA with Tukey's post-hoc test (n = 3/group, biological replicates). Data are presented as mean +/- SEM (*p < 0.05, **p < 0.01, ****p < 0.0001).

The following description was added in the text:
MHC-I expression on both dendritic and cancer cells is essential for the effector function (Figure EV3, H).

Comment #10: In addition, in Figure 4J, I was surprised that EVs + macbecin II treatment exhibit increased MHC-I expression compared to macbecin II treatment only. Do the authors have an explanation for this?

Response: We have shown that the combination of macbecin II and the vaccine promoted the infiltration of immune cells, such as CD4 and CD8 T cells, which are known to enhance MHC expression on the cell surface through cytokines like IFN- γ and TNF- α (Sunden *et al*, 2010; Zhou, 2009). Additionally, we previously demonstrated that the vaccine decreases Treg cells, which are known to suppress MHC-I expression (Zhang *et al*, 2024). Similarly, NK cells, which are enriched by the vaccine and can present antigens independently of antigen-presenting cells

(Wu *et al*, 2023), are known to upregulate MHC-I expression through the secretion of IFN- γ (Dunn *et al*, 2006).

Comment #11: In Figure 5, does macbecin II also increase/stabilise MHC-II expression and immunoregulatory receptors (PD-1, etc...)? I wonder whether there is a widespread effect of macbecin II, especially given its mode of action, which could also contribute to its effect in vivo.

Response: We thank the reviewer for this concern. We have examined the expression of MHC-II and found that macbecin II also promoted the stability of MHC-II. Additionally, we examined the PD-L1 expression on tumor cells on macbecin II treatment and found that there was no significant change in PD-L1 expression, demonstrating that macbecin II does not have widespread effect on other cell surface receptors. These results are incorporated in the **revised Appendix Figure S1, D-F and J,K**.

Appendix Figure S1: Macbecin II upregulates MHC-I and II through inhibition of lysosomal degradation.

D, E: DCIS.com and MCF10CA1a cells were seeded in 24-well plates and treated with the indicated doses of macbecin II for 48 hours. Total RNA was isolated, and cDNA synthesis was performed using the iScript™ cDNA Synthesis Kit according to the manufacturer's instructions, followed by SYBR Green-based real-time PCR. GAPDH was used as the internal control. Statistical analysis was performed using the one-way ANOVA with a Tukey post-hoc test (n = 3/group, biological replicates). **F:** DCIS.com cells were treated with cycloheximide (50 μ g/ml) and macbecin II (0.5 μ M) for the indicated times. MHC-II expression was examined by Western blot (left panel) which was quantified using ImageJ (right panel) (n = 3/group, biological replicates). A two-tailed unpaired Student's t-test was used for analysis. Data are represented as mean \pm SEM (*p < 0.05, **p < 0.01, ****p < 0.0001).

Appendix Figure S1: Macbecin II upregulates MHC-I and II through inhibition of lysosomal degradation.

J: DCIS.com cells were seeded in a 24-well plate and treated with the indicated doses of macbecin II for 48 hours. Total RNA was isolated, and cDNA synthesis was performed using the iScript™ cDNA Synthesis Kit according to the manufacturer's instructions, followed by SYBR Green-based Real-Time PCR. GAPDH was used as the internal control. Statistical analysis was performed using the one-way ANOVA with a Tukey post-hoc test ($n = 3/\text{group}$, biological replicates). **K:** DCIS.com cells were treated with macbecin II ($0.5 \mu\text{M}$), and PD-L1 expression was examined by western blot.

Accordingly, the following description was added in the text:

We also examined the expression of MHC-II and found that macbecin II did not regulate the RNA expression of MHC-II (Appendix Figure S1,D ,E), but it promoted the stability of MHC-II protein (Appendix Figure S1,F). Similar to MHC-I, we found that inhibiting proteasomal degradation did not affect the stability of MHC-II (Appendix Figure S1,G), while the lysosomal degradation inhibitor bafilomycin stabilized MHC-II expression (Appendix Figure S1, H). Additionally, the total expression of MHC-II protein was rescued when treated with C381 and macbecin II (Appendix Figure S1, I). We examined PD-L1 expression on tumor cells following macbecin II treatment and found no significant change in PD-L1 expression. This demonstrates that macbecin II does not have widespread effects on other cell surface receptors (Appendix Figure S1, J,K).

References:

- Dunn GP, Koebel CM, Schreiber RD (2006) Interferons, immunity and cancer immunoediting. *Nat Rev Immunol* 6: 836-848
- Peter I, Mezzacasa A, LeDonne P, Dummer R, Hemmi S (2001) Comparative analysis of immunocritical melanoma markers in the mouse melanoma cell lines B16, K1735 and S91-M3. *Melanoma Res* 11: 21-30
- Seliger B, Wollscheid U, Momburg F, Blankenstein T, Huber C (2001) Characterization of the major histocompatibility complex class I deficiencies in B16 melanoma cells. *Cancer Res* 61: 1095-1099
- Sunden Y, Yano S, Ishida S, Ochiai K, Umemura T (2010) Intracerebral vaccination suppresses the spread of rabies virus in the mouse brain. *Microbes Infect* 12: 1163-1169
- Wu K, Lyu F, Wu SY, Sharma S, Deshpande RP, Tyagi A, Zhao D, Xing F, Singh R, Watabe K (2023) Engineering an active immunotherapy for personalized cancer treatment and prevention of recurrence. *Sci Adv* 9: eade0625
- Zhang D, Zhan D, Zhang R, Sun Y, Duan C, Yang J, Wei J, Li X, Lu Y, Lai X (2024) Treg-derived TGF- β 1 dampens cGAS-STING signaling to downregulate the expression of class I MHC complex in multiple myeloma. *Sci Rep* 14: 11593
- Zhou F (2009) Molecular mechanisms of IFN-gamma to up-regulate MHC class I antigen processing and presentation. *Int Rev Immunol* 28: 239-260

12th Feb 2025

Dear Dr. Watabe,

Thank you for submitting your revised study. We have now received the reports from the referees who evaluated your revised manuscript. As you will see from the reports below, they are satisfied with the revisions, and I will therefore be able to accept your manuscript once the following editorial issues are addressed:

1/ Referees' comments:

Please consider the comment of referee #2 regarding the title.

2/ Manuscript text:

- Please remove the underlined text and only keep in track changes mode any new modification.
- Please remove the figures from the manuscript file and add the EV figure legends. Please correct the order of the manuscript sections as follows: Abstract, Keywords, Introduction, Results, Discussion, Methods, Acknowledgements, Disclosure and competing interests statement, References, Figure legends, Expanded View Figure legends .
- Authors: there is a discrepancy between Ravindra Pramod Deshpande in the manuscript text and Ravindra Deshpande in the submission system, please correct.
- We can accommodate a maximum of 5 keywords, please adjust accordingly.
- Methods and Protocols should be renamed "Methods":
 - o Thank you for providing a Reagents and tools table. Please remove it from the manuscript file and upload it as a separate file.
 - o Cells: please indicate whether the cells were authenticated and tested for mycoplasma contamination.
 - o Mice: please indicate the origin of the animals.
 - o Statistical analysis: please provide a statement on sample size, exclusion/inclusion criteria, blinding and randomization.
- Data Availability: It is mandatory to include a 'Data Availability' section after the Methods. Before submitting your revision, primary datasets produced in this study need to be deposited in an appropriate public database, and the accession numbers and database listed under 'Data Availability'. In case you have no data that requires deposition in a public database, please state so in this section ("This study includes no data deposited in external repositories"). Note that the Data Availability Section is restricted to new primary data that are part of this study.
- Funding information should be part of Acknowledgements. The information provided in the manuscript and the submission system should match, please adjust accordingly. The text from the Comments box needs to be removed and the funder and grant need to be provided as a separate entry.
- Please rename "Competing interest statement" to "Disclosure statement and competing interests". Please review our updated policy <https://www.embopress.org/competing-interests> and update your competing interests if necessary.
- Author contributions: CRediT has replaced the traditional author contributions section because it offers a systematic machine readable author contributions format that allows for more effective research assessment. Please remove the Authors Contributions from the manuscript and use the free text boxes beneath each contributing author's name in our system to add specific details on the author's contribution. More information is available in our guide to authors.
- Please remove "Patient consent for publication: Not applicable"
- Ethics approval should be in the Methods section.

3/ Figures and Appendix:

- Figures 4 and EV4 spread on two pages, which is not permitted. Please adjust accordingly.
- The file with EV figures and legends is not needed as we need EV figures uploaded separately and their legends in the manuscript.
- We would suggest making your Appendix figure an EV figure (Figure EV6).
- Please make sure that all figures and figure panels are referenced in the text. Please note that a Table EV1 is called out, but it is missing.
- Please address the queries from our copy editors in the figure legends:
 1. Please note that the exact p values are not provided in the legends of figures 1C-E; 2A-D; 3B, C, F; 4C-I, L-T, 5B-H; 6C, D, F; EV1 A-D; EV2 A-F; EV3 C, D, E, G, H; EV4 B, C, D, E, F, G, H, I, J;
 2. Please indicate the statistical test used for data analysis in the legend of figure EV3 C, EV4 J
 3. Please note that in figures 1D, E; 2A-D; 4L-N; 5E-G; 6C; EV1C, D; EV2 A, B, D-F; EV3D, E, G; EV4 I, J; there is a mismatch between the annotated p values in the figure legend and the annotated p values in the figure file that should be corrected.
 4. Please note that the p value is not represented in the figure 5I, 6E, however statistical test related information is provided in the legend of the corresponding figure. This needs to be rectified.

4/ Thank you for providing Source Data. Please reorganize the files into separate zip folders - one folder per figure where each folder would have separate folders/files - one per panel. Please complete the SD checklist.

5/ Checklist:

Please make sure that the right column is also filled appropriately (i.e. cell materials, experimental animals).
Please fill in the subsection on inclusion/exclusion criteria in the 'Experimental study design and statistics'.

6/ Synopsis: Please also provide a stand first (maximum 300 characters including spaces) and upload the text (with the bullet points) as an individual word document.

Thank you for providing a nice visual abstract. Please upload it as a jpeg, TIFF or png file, 550 px wide x 300-600 px high. Please make sure that the text remains legible. A cropped portion of this image will serve as thumbnail for the table of content on our webpage.

7/ We note that you agree with the publication of the RPF, which will include the anonymous referee reports, your point-by-point response and all pertinent correspondence relating to the manuscript. The Authors checklist will be published at the end of the RPF.

I look forward to receiving your revised manuscript.

Yours sincerely,

Lise Roth

***** Reviewer's comments *****

Referee #1 (Remarks for Author):

The authors have satisfactorily addressed my various concerns through novel additional experiments.

Referee #2 (Comments on Novelty/Model System for Author):

The authors carried out many novel experiments, and they successfully addressed my concerns. They characterized e.g. the EV preparates, they included a mouse model for the endogeneous formation of mammary tumors, and they also used another model instead of B16 melanoma cells. In my opinion, the manuscript can be accepted for publication. However, since the manuscript focuses on the effect of macbecin II, I would suggest to include it into the title of the manuscript, e.g. "MHC-I upregulation BY MACBECIN II in the solid tumors potentiates the effect of active immunotherapy".

Referee #2 (Remarks for Author):

Many thanks to the authors for carrying out novel experiments, they have now successfully addressed all my concerns. However, since the manuscript focuses on the effect of macbecin II, I would suggest to include it into the title of the manuscript, e.g. "MHC-I upregulation by macbecin II in the solid tumors potentiates the effect of active immunotherapy".

Referee #3 (Remarks for Author):

The new controls and experiments strengthen the conclusions and overall manuscript. I don't have any further concerns.

All editorial and formatting issues were resolved by the authors.

21st Feb 2025

Dear Dr. Watabe,

Thank you for submitting your revised study. I have gone through all the files, and a few editorial points remain to be addressed before I can accept your manuscript:

- Several figure panels appear to be distorted or stretched, which could be due to a problem during transfer, or file compression. Please carefully check each panel (in particular, Fig. 4L, Fig. 5B, Fig. 6C).
- Please carefully check Fig. 4Q and 4R, which appear identical (as well as their Source Data).
- If possible, please provide Source Data for Fig. 4E, 4N, 5D, 5E, 5F
- Figure legends: Please indicate the statistical test used for data analysis in the legend of figure EV3 C, EV4 J.
- Author Checklist: please fill in the right column where appropriate (i.e. cell materials, experimental animals). Also kindly fill in the subsection on blinding in the 'Experimental study design and statistics'.
- Synopsis: I introduced minor edits in your text, let me know if you agree with the following:

"Macbecin II was identified as a compound that upregulates MHC-I expression, enhancing immune-mediated cancer cell killing. In animal models, combining macbecin II with immunotherapies, including vaccines and checkpoint inhibitors, improved treatment efficacy.

1. Macbecin II upregulated MHC-I expression and enhanced antigen-dependent killing of cancer cells
2. Active immunotherapy in the form of IL2-ep13nsEV synergized with macbecin II and reduced tumor growth and mitigated metastasis.
3. Enhanced MHC expression by macbecin II potentiated the effect of anti PD-1 immune checkpoint blockade."

- Thank you for providing a visual abstract. The image seems to be slightly stretched, could you please kindly check and send a new image?

Thank you for bearing with these last changes.

Looking forward to receiving your revised files.

Yours sincerely,

Lise Roth

All editorial and formatting issues were resolved by the authors.

28th Feb 2025

Dear Dr. Watabe,

Thank you for submitting your revised files. I am pleased to inform you that your manuscript is accepted for publication and is now being sent to our publisher to be included in the next available issue of EMBO Molecular Medicine.

Yours sincerely,

Lise Roth
